**Using kinetic energy measurements from altimetry to detect shifts in the**
**positions of fronts in the Southern Ocean**
Don P. Chambers[1]
[1] College of Marine Science, University of South Florida, St. Petersburg, FL
Correspondence to: D. Chambers (donc@usf.edu)
**Abstract.** A novel analysis is performed utilizing cross-track kinetic energy (CKE) computed
from along-track sea surface height anomalies. The mid-point of enhanced kinetic energy
averaged over three-year periods from 1993 to 2016 is determined across the Southern Ocean
and examined to detect shifts in frontal positions, based on previous observations that kinetic
energy is high around fronts in the Antarctic Circumpolar Current system due to jet instabilities.
It is demonstrated that although the CKE does not represent the full eddy kinetic energy
(computed from crossovers), the shape of the enhanced regions along groundtracks is the same,
and CKE has a much finer spatial sampling of 6.9 km. Results indicate no significant shift in the
front positions across the Southern Ocean, on average, although there are some localized, large
movements. This is consistent with other studies utilizing sea surface temperature gradients, the
latitude of mean transport, and probability of jet occurrence, but is inconsistent with studies
utilizing the movement of contours of dynamic topography.

## 1. INTRODUCTION

There is as much we don't know about the circulation of the Southern Ocean as we do. Although the current system is routinely called the Antarctic Circumpolar Current (ACC), it consists of several fronts with distinct water properties to the north and south of the fronts (Nowlin and Clifford, 1982; Orsi et al., 1995; Belkin and Gordon, 1996). The most significant of these fronts, responsible for the majority of the ACC volume transport (e.g., Cunningham et al., 2003), are the Subantarctic Front (SAF) and the Polar Front (PF). However, even this is not a realistic picture of the circulation in the Southern Ocean, since at any specific time, there can be from three to ten narrow jets around the fronts that are highly variable in strength and location, masking the specific frontal boundary (Sokolov and Rintoul, 2007, 2009a, 2009b; Sallee et al., 2008; Thompson et al., 2010; Thompson and Richards, 2011; Langlais et al., 2011; Graham et al., 2012; Chapman, 2014; Gille, 2014; Kim and Orsi, 2014; Shao et al., 2015; Chapman, 2017a). Although positions of fronts have been estimated throughout the Southern Ocean, primarily using gradients of subsurface density measured from hydrographic sections (Orsi et al., 1995), contours of dynamic topography (Sokolov and Rintoul 2007, 2009a, 2009b; Langlais et al., 2011), or a combination Kim and Orsi (2014), in many places there are no strong currents that can be measured near the front position (Chapman, 2014; 2017a).

Because of the highly variable nature of jets and the lack of clear observational detection of fronts in some areas, the literature has become muddled over the difference between a front and a jet, primarily because the "front" is rarely observed at any specific time due to the high-variability of jets (Thompson et al., 2010; Thompson and Richards, 2011; Chapman 2014; 2017a). However, even in the presence of highly variable jets, methods have been developed to determine mean fronts positions in a probabilistic sense. Thompson et al. (2010) demonstrated

one could define fronts in the Southern Ocean by computing probability density functions of
potential vorticity in an eddy-resolving general ocean circulation model. Chapman (2014, 2017a)
later showed this could also be done using localized gradients in dynamic topography (i.e., high
geostrophic velocity) using satellite altimeter observations, but again, only as statistical
probability. This is because these areas of enhanced gradients and velocity are more reflective of
jets, which strengthen and die, appear and disappear, bifurcate and join back together. Because of
this, they can only be detected on average 10-15% of the time. However, Chapman (2014,
2017a) has demonstrated that, at least in a mean sense, fronts defined by mean dynamic
topography contours (commonly known as the "contour method") do lie within the probability
distribution inferred from "gradient" methods.
An open question is how the fronts and jets that comprise the ACC will respond in a
warming climate. Analysis of climate models (which cannot simulate jets in the Southern Ocean)
suggests that as the atmosphere warms, the winds that drive the fronts and jets of the ACC will
migrate south (e.g., Fyfe and Saenko, 2006; Swart and Fyfe, 2012). It should be noted, however,
that the mean position of the southern hemisphere westerlies in the models lies significantly
equatorward of the true position (e.g., Figure 2 in Fyfe and Saenko, 2006). Thus, it is not entirely
clear whether the model is predicting a true shift in the wind position, or whether the model has
not yet reached equilibrium with winds in the proper location.
Still, based on these model results, researchers have been testing the hypothesis that as winds
in the Southern Ocean shift south, the frontal positions and jets will also migrate south. So far,
the results are mixed. Using the contour method and tracking how the dynamic topography
contours associated with a front position shift in time, Sokolov and Rintoul (2009b) found that
the SAF and PF had both moved south by approximately 60 km over 15 years between 1993 and
2008. Kim and Orsi (2014) recently updated this analysis and found that while the average
frontal position across the Southern Ocean indicates a strong southward shift, this is due
primarily to substantial shifts only in the Indian Ocean sector. They found no significant shifts
throughout the Pacific or Atlantic Ocean sectors using the contour method.
The primary assumption of these analyses is that if a contour of dynamic topography shifts
south, it is uniquely caused by a front moving south. This is not necessarily true. Gille (2014)
recently demonstrated that all contours in the Southern Ocean have shifted south on average, and
that this follows from the observed rise in sea level – as the sea surface height rises, the contours
will appear to shift south. While this breaks down at the far south and north of the ACC where
dynamic topography gradients are small, these areas are far away from the PF and SAF and so
have not been considered in previous analyses. Gille (2014) used a different measure to
determine the position of the ACC fronts, based on the latitude of the mean surface transport of
the ACC measured by altimetry, which is in essence a mean location of all the jets in the
Southern Ocean. She found no significant shift on average, but considerable interannual
variability, especially regionally.
Another factor other than sea level rise can cause the dynamic topography contour to shift
south -- if the magnitude and width of the jet has changed. This is demonstrated in **Figure 1**,
where we show the mean dynamic topography from two jet scenarios: 1) where the peak of two
Gaussian shaped jets have shifted south, and 2) where the peak has not shifted, but the magnitude
has decreased, the width has broadened, and the shape has become slightly skewed. Although the
resulting topography profiles are not identical, they are similar, and both suggest a southward
movement of dynamic topography contours.

Researchers using other methods also find little or no southern migration of the fronts or jets

in the Southern Ocean as a whole. Graham et al. (2012) used a high-resolution model to show
that the Polar Front and Subantarctic Front are constrained by bathymetry, even in increasing and
shifting winds. Shao et al. (2015) utilized the skewness of sea level anomalies to identify front
positions, and found no southward motion, but did find changes in the east Pacific correlated
with the Southern Annual Mode. Chapman (2017a), using positions of fronts determined from
the probability of jet locations, also found no significant southward movement, but high
interannual variability. Finally, Freeman and Lovenduski (2016a) used weekly estimates of the
Polar Front position determined from satellite sea surface temperature (SST) gradients to show
no significant southward shift between 2002 and 2014 on average, except in the Indian Ocean.
They also found a statistically significant northward shift of the PF in part of the south Pacific.

Thus, recent studies all agree that the Subantarctic Front and Polar Front have not shifted

south, even though there is evidence the winds have shifted south in the austral summer months
(Swart and Fyfe, 2012). It should be noted that when averaged over the full calendar year,
however, there has been no significant shift in the wind position (Swart and Fyfe, 2012).

In this paper, we develop a new method to study linear shifts in the position of the fronts in

the Southern Ocean, based on tracking the location of envelopes of kinetic energy measured by
satellite altimetry. It is known from modeling studies that the front positions are associated with
increased kinetic energy, due to instabilities in the jets and interactions with bathymetry
(Thompson et al., 2010; Thompson and Richards, 2011). After demonstrating that kinetic energy
computed from along-track satellite altimetry forms relatively wide envelopes of enhanced
energy that occur within the probability range of jets and fronts (e.g., Chapman, 2017a), we track
the positions of these envelopes from 1993 until 2016 to quantify if the envelopes have shifted
south by a statistically significant amount. Since kinetic energy is highest around fronts in the
Southern Ocean (e.g., Thompson et al., 2010; Thompson and Richards, 2011; Chapman, 2017) ,
it follows that if the fronts have shifted south, then the envelope of high kinetic energy should
also move by a comparable amount. We do not purport that our method derives the actual
position of either a front or a jet due to the relatively wide swath of enhanced kinetic energy on
either side of fronts related to variability of jets. Instead, we only purport that it can indicate
shifts in the frontal position, because if a front has shifted south by 100 km (for instance), then
the band of enhanced kinetic energy should also shift south by a comparable amount. It is
difficult to reconcile a frontal shift without a displacement of kinetic energy.
Since the kinetic energy calculation is based on estimating gradients of sea level anomalies,
this approach is similar to other gradient methods for detecting fronts or jets (e.g., Chapman,
2014; 2017a; Gille, 2014; Freeman and Lovenduski, 2016a). It differs from these approaches,
however, in that instead of determining individual gradients and tracking these over time, it looks
for regions of high gradients (i.e., high energy) surround by regions of low gradient (i.e., low
energy). This allows us to detect envelopes for every time-period considered, instead of only a
fraction of the time, allowing for better tracking of the change over time.
Section 2 will describe the data and methods used, while section 3 will present results,
including evaluation of the method for detecting mean positions of fronts and for tracking their
change over time. Section 4 will discuss the results in the context of previous studies and
evaluate the usefulness of the method.
**2. DATA AND METHODS**
We utilize geostrophic surface current anomalies computed from the 24-year record of 1-Hz
sea surface height (SSH) data along the TOPEX/Poseidon (T/P) groundtrack in the Southern
Ocean (**Figure 2**). The altimetry data used are from four separate altimeter missions:
TOPEX/Poseidon (January 1993 – January 2002), Jason-1 (February 2002 – July 2008), Jason-2
(August 2008 – August 2016), and Jason-3 (August 2016 – December 2016). Because the
official TOPEX/Poseidon (T/P) geophysical data records (GDRs) have not been updated since
the late 1990s, we utilize the corrected data products from the Integrated Multi-Mission Ocean
Altimeter Data for Climate Research provided by Beckley et al. (2010) at the NASA PO.DAAC
site (https://podaac.jpl.nasa.gov/Integrated_Multi-Mission_Ocean_AltimeterData). Jason-1 data
are from the GDR-C version and were downloaded from the NASA PO.DAAC site in June 2010.
Jason-2 are from the GDR-D version and were downloaded from NOAA NODC
(ftp://ftp.nodc.noaa.gov/pub/data.nodc/jason2) between August 2012 and June 2016. Jason-3 are
also from the GDR-D version and were downloaded from NOAA NODC
(ftp://ftp.nodc.noaa.gov/pub/data.nodc/jason3) on August 7 and 8, 2017.

We utilize the 1-Hz along-track SSH data from the four altimeters and compute sea level

anomalies by interpolating the DTU10 mean sea surface model (Andersen and Knudsen, 2009;
http://www.space.dtu.dk/english/Research/Scientific_data_and_models/downloaddata) to the
SSH location using bilinear interpolation. The DTU10 mean sea surface model is based on SSH
from multiple altimeters averaged over 17 years in a rigorous and consistent manner (Andersen
and Knudsen, 2009). T/P, Jason-1, and Jason-2 data were all included.  All recommended
geophysical and surface corrections (e.g., water vapor, ionosphere, sea state bias, ocean tides,
inverted barometer, etc) have been applied, to correct for biases introduced by atmospheric
signal refraction and sea state effects (e.g., Chelton et al., 2001).

We utilize this record rather than the gridded products based on mapping SSH from multiple

altimeters (e.g., Ducet et al., 2000; Pujol et al., 2016), because the along-track data have a finer
resolution in space (6.9 km along the groundtrack) and we recently demonstrated that the
mapped altimetry data underestimated eddy kinetic energy (EKE) throughout the Southern
Ocean compared to using along-track data by as much as 60-70% (Hogg et al., 2015).  While the
along-track sea level anomalies are filtered to reduce noise and thus may attenuate some signal,
the filtering used (described later in this section), is less than that used for the mapped data,
which uses observations from as long as 20 days and 200 km away to influence the mapped
value. By filtering only alongtrack data, the time differences are small (a few minutes at most),
and the spatial influence is less than 100 km. Tests with unfiltered data accounting for estimated
random noise in the sea level anomaly data suggests attenuation of kinetic energy is minimal
with this approach and, more importantly, that the shape of the kinetic energy envelope does not
significantly change.

One can only compute EKE from alongtrack data at crossover points, where the ascending

and descending groundtracks cross (Figure 2). Knowing the groundtrack angle with the north
meridian ($\theta$) one can compute the zonal ($d\eta/dy$) and meridional gradients ($d\eta/dx$) of SSHA
directly from the gradients of SSHA for the ascending pass ($d\eta/dr_{asc}$) and descending pass
($d\eta/dr_{des}$) using simple geometry (Parke et al., 1987)
$$\frac{d\eta}{dy} = \frac{\left[\frac{d\eta}{dr_{asc}} - \frac{d\eta}{dr_{des}}\right]}{2\sin\theta}, \frac{d\eta}{dx} = \frac{\left[\frac{d\eta}{dr_{asc}} + \frac{d\eta}{dr_{des}}\right]}{2\cos\theta}, \tag{1}$$

noting that this formulation assumes the gradients represent the derivative of the northern SSHA
relative to the southern SSHA (for both the ascending and descending passes). Once this is
computed, the velocities can be computed directly from the zonal and meridional gradients:
$$u = -\frac{g}{f}\frac{d\eta}{dy}, \ v = \frac{g}{f}\frac{d\eta}{dx},$$
(2)

where $g$ is the acceleration due to gravity, and $f$ is the Coriolis parameter
This formulation assumes that the velocity field has not changed significantly between the
times the two passes fly over the crossover point. At high latitudes, the majority of crossovers (>
78%) have a time separation of less than 3 days. At 40°S, the average propagation speed of an
eddy is about 3 cm s$^{-1}$ [Chelton et al., 2007], meaning the eddy would have only been displaced
by 8 km at most over this period. At higher latitudes, this is even less. Considering the diameter
of eddies at these latitudes are of order 100 km [Chelton et al., 2007], the movement is not large
enough to cause a significant change in velocity at the point. The primary problem with
velocities computed from crossovers is the smaller number compared to using gridded data, or
the time-varying, anomalous geostrophic current normal to the groundtrack ($u_T$). This can be
computed directly from the derivative of the SSH anomaly ($\eta$) along the ground-track distance
($dr$) from
$$u_T = -\frac{g}{f}\frac{d\eta}{dr}$$
(3)

This cross-track current is a projection of both the zonal ($u$) and meridional ($v$) components of
the full anomalous velocity field. However, neither $u$ nor $v$ can be determined unambiguously
from $u_T$. Here, we merely examine the variability of $u_T$ without making any assumptions
concerning how it may be related to the full velocity, or $u$ and $v$.
Because derivatives of SSHA (Equations 1 and 3) have to be computed numerically (here,
center-differences are used) and $\eta$ contains significant noise at the 1 Hz sampling-rate of the
altimeters, we optimally interpolate $\eta$ along-track using a model of the covariance of the signal
and error. We used the method of Wunsch (2006, Chapter 3) and a covariance function modeled
as a Gaussian with a roll-off of 98 km and random noise of 2 cm, which was determined from the
autocovariance of all TOPEX/Poseidon, Jason-1, and Jason-2 SSHA data from 1993-2015
between 40°S and 65°S.
Once $u_T(t)$ was computed at each 1-sec bin along the groundtracks in Figure 2 for each 10-
day repeat cycle, the cross-track kinetic energy (CKE) was computed as $CKE(x,t) = 0.5\ u_T(x,t)^2$,
where $x$ here is used to denote a generic 1-sec bin along the ground track.  We also computed the
full EKE at the more limited crossover points as $EKE(x,t) = 0.5(u(x,t)^2 + v(x,t)^2)$.
The CKE values were averaged over the entire 24-year record and examined for each
groundtrack segment (both ascending and descending) to judge where CKE was exceptionally
high (Figure 3). We also computed CKE using the raw values of $\eta$ with no optimal interpolation
and compared to that computed with optimal interpolation. The locations of high CKE were the
same, although values were significantly higher with the unsmoothed data. The quiescent regions
of the ocean also showed considerably more noise, making it more difficult to determine
boundaries of elevated CKE. For this reason, the values determined from the optimally
interpolated data were used.
Several criteria were utilized to quantify where the high CKE values were considered to be
associated with fronts. First, we constrained the southern boundary to be 5° south of the Orsi et
al. (1995) values of the PF and the northern boundary to be 5° north of the SAF. Secondly, we
used a lower-limit for CKE of 200 $cm^2\ s^{-2}$ for detection and tested that the width of the envelope
of high CKE above the lower-limit was at least 100 km. The requirement that the envelope be
greater than 100 km was done to reduce the impact of eddies in an otherwise quiescent region,
since the diameter of eddies in the Southern Ocean is about 100 km. The CKE lower-limit was
determined via iteration with different limits. For each case, the average center of the CKE
envelope averaged over 24-years (based on the mean of the first and last points to exceed the
lower-limit) was computed and compared visually to the Orsi et al. (1995) front positions.  200
$cm^2 s^{-2}$ was selected because there were a significant amount of CKE envelope centers clustered
around the Orsi et al. (1995) fronts and the envelopes were found for every 10-day repeat cycle.
Using a higher limit resulted in fewer detections, especially when smaller time-averages were
used. Using a lower limit, we could find more potential front positions based on CKE, but many
were far from the front positions estimated by Orsi et al (1995) and other authors (e.g., Kim and
Orsi, 2014; Freeman and Lovenduski, 2016a; Chapman, 2017).

An example of a detected high CKE envelope is shown in Figure 3, based on the average of

CKE between 1993 and 2015 computed from T/P-Jason satellite pass 207 in the south Indian
Ocean. This pass starts at 64.3°S near the prime meridian and extends to 41.2°S and 41°E
longitude. There is clearly a wide envelope of enhanced CKE greater than 200 $cm^2 s^{-2}$ between
55°S and 47°S.

The mean CKE profile pictured in Figure 3 has multiple local maxima, most likely associated

with variability of the narrow jets that surround the front. They may also represent two separate
fronts (and frontal-related jets) that are close in space. Some frontal climatologies find the SAF
and PF are separated by fewer than 100 km in the South Indian Ocean (between 30°E and 40°E),
the South Pacific (between 220°E and 230°E), and the South Atlantic (310°E and 330°E) (Figure
2). CKE computed in these areas may encompass energy around both fronts. However, if the
fronts have both shifted south (as reported in some studies), then CKE should also shift south
and so tracking CKE should observe the shifts in frontal location.

Figure 4 shows the behavior of CKE along this pass for different 3-year periods. Note that

the number of clearly defined maxima ranges from a low of 4 for the 2014-2016 average to 9 in
1993-1995. Note that even with a fixed and stationary front, there may be highly variable
locations of peaks in CKE around the front, due to the meandering and disappearance/formation
of jets (e.g., Chapman, 2017a). Thus, tracking the specific jet locations is not an optimal method
of tracking frontal shifts. While other studies have estimated positions of these maxima in SSHA
gradients on daily intervals (e.g., Chapman, 2017a), one does not obtain a consistent number of
maxima each time, making the determination of shifts difficult. Moreover, note that although
there are two general peaks in CKE in the long-term mean profile, the minimum between them is
still higher than 200 cm$^2$ s$^{-2}$. A minimum is also not well defined in several of the shorter
averaging periods (for example, 2008-2010).

Thus, instead of attempting to track all the maxima of CKE individually – analogous to

tracking steepest gradients, as in Thompson et al. (2010), Graham et al. (2012), or Chapman
(2017a) – we track an estimate of the center of the envelope of enhanced CKE, as it exists in all
averaging periods. The assumption we make in doing this is that the localized maxima are
associated with variable jets, but the position of the envelope of high CKE is related to the
general position of the front, and that if the front has systematically shifted then the CKE
envelope will have shifted as well. Other studies have tracked the mean latitude of the integrated
transport computed between dynamic height contours that are picked to represent the southern
boundary and the northern boundary that encompass all the fronts in the ACC (Gille, 2014). One
issue with this approach is how to uniquely determine the northern and southern boundary
contours without potentially biasing the result (e.g., using a priori fixed boundaries and ignoring
that they might have shifted). The method we propose will determine the boundaries of the
integration uniquely for each pass based solely on the level of CKE relative to the peak of the
enhanced CKE envelope. Moreover, it allows for two or more distinct CKE envelopes along
each pass (i.e., related to different fronts), whereas the Gille (2014) method can only compute
one mean latitude for all fronts in the between the prescribed southern and northern boundaries.
Thus, our method is more flexible in determining boundaries around any particular front,
provided the orientation of the groundtrack is such that the majority of jets are perpendicular to
it.

There are many different ways to compute a "center" of the envelope, ranging from the

average of the two end points, to a centroid calculation, to computing the point where the integral
of CKE over distance is balanced on both sides, which we call the "half-power point." We have
selected the latter to use, as it defines a "center" closer to the peak of CKE in the envelope. This
is advantageous when the CKE curve is slightly skewed, with less magnitude on one side and
more on the other. Assuming that the variability (and hence CKE) would be highest near the
front (i.e., what is assumed in studies using the gradient method), finding a center of the
envelope that is biased toward peak CKE is a reasonable approach.

The half-power point ($x_{mid}$) is computed so that

$$\int_{x_{south}}^{x_{mid}} CKE(x)\,dx = \frac{1}{2}\int_{x_{south}}^{x_{north}} CKE(x)\,dx\,, \qquad (4)$$

where $x_{south}$ and $x_{north}$ are computed by first finding the maximum of CKE in the envelope above
200 cm$^2$ s$^{-2}$, then finding the first value to the north just below 25% of that peak along with the
similar value to the south (shown in Figure 3). Values other than 25% of the peak were tested.
Using value greater than this, up to 50%, resulted in no significant difference in the half-power
point. Using values smaller resulted in some boundaries not being defined. Thus, 25% of peak
CKE was considered reasonable.  If multiple regions of enhanced CKE were found along the
same track, this process was carried out for each of them. This was done for all the 24-year mean
CKE profiles to establish the mean locations of the fronts between 1993 and 2016.

A similar procedure was done for CKE averaged over discrete 3-year intervals, starting in

January 1993 and ending in December 2016. A 3-year average was used to reduce the influence
of individual eddies on determining the envelope, and to reduce interannual variations in the
front position, which have been observed in other studies at some locations (e.g., Kim and Orsi,
2014; Shao et al., 2015). In particular, Kim and Orsi (2014) and Shao et al. (2015) found
significant correlation with the Southern Annular Mode, which has a quasi-biennial oscillation
(Hibbert et al., 2010). By averaging over three years, we found 8 distinct, statistically
uncorrelated samples of CKE for each groundtrack from which to deduce shifts in the half-power
point. We tested different averaging periods (ranging from 1- to 4-years), but found the estimate
in overall shift of the half-power point over the 24-year period was insensitive to the choice.

## 3. RESULTS AND ANALYSIS

The first thing tested was how well CKE represented the full EKE. If CKE does not have the

same general shape as EKE, then using it as a proxy for EKE to determine high energy envelopes
is not valid. After finding satellite passes with high CKE as discussed in Section 2, EKE was
computed along the same pass, using the crossover method (Equations 1 and 2).

Although CKE is lower than EKE along all groundtracks (see Figure 5 for examples), the

pattern of KE rise then fall is virtually identical. CKE, however, has the benefit of higher and
more regular sampling. Thus, we conclude CKE is a reasonable proxy for locating front positions
even though it may not be useful for quantifying the full energy of the anomalous currents.
Four general types of enhanced CKE were found (Figures 4 and 5). In most regions, the
envelope in CKE is more or less symmetrical (52% of cases). Only a few profiles have two
distinct regions of enhanced CKE that were identified, with a clearly defined minimum below
200 cm$^2$ s$^{-2}$ between them in all time periods (3% of cases). 20% of the passes have multiple
peaks that vary in time but have no consistent minimum between the peaks (i e., Figure 4), while
25% have a skewed envelope (Figure 5), with a long rise in CKE followed by a sharp drop-off.
In all cases, though, the shape of the CKE envelope closely follows that of EKE, although the
amplitude was attenuated, by anywhere from 25-50%. Having closer samples of CKE, however,
allows for a better computation of the half-power point and possible shifts.
Figure 6 shows the locations of the half-power points determined from the mean CKE
profiles, along with estimate of the front position based on different methods: density gradients
from historical hydrographic sections (Orsi et al., 1995), dynamic topography contours (Kim and
Orsi, 2014), and the gradient of sea surface temperature (Freeman and Lovenduski, 2016a).
There are two estimates of the SAF and SACCF, and three of the PF. One of the PF estimates
(from Freeman and Lovenduski, 2016a) includes the standard deviation of the daily estimates.
It is important to note the large differences in estimates for the same front, which indicates
how difficult it is to determine fronts in a highly variable current system like the ACC. For
instance, in the Indian Ocean at 50°E, Freeman and Lovenduski (2016a) find the PF at the same
location that Orsi et al. (1995) found the SAF, while Kim and Orsi (2014) find it significantly
farther south. The SAF determination using the contour method (Kim and Orsi, 2014) is
substantially farther north than the one determined from hydrographic data (Orsi et al., 1995) at
most longitudes. These differences are likely due to differences in the time-span, differences in
methodologies, and uncertainty in the data utilized. All lead to a level of uncertainty in the
determination of a specific front at any time.

The half-power points of enhanced CKE generally occur near or between the fronts estimated

by different methods (i.e., the three different PF estimates), indicating they are at least within the
uncertainty bounds of frontal detection by other methods. Some values are at locations either
north or south of the other front estimates by as much as 3°, but it should be noted that the
standard deviation of the PF estimated by Freeman and Lovenduski (2016a,b) averages 2-3°.
Using a PF variability statistic as an indicator of variability of all fronts, one can conclude the
location CKE half-power points are well within the level of expected frontal variability and so
not statistically too distant from a front location.

One may question whether the relatively wide envelopes of enhanced CKE overlap more

than one front. This is a possibility, but if both fronts have moved south as some have argued
(e.g., Sokolov and Rintoul, 2009b), then the CKE envelope should also shift, regardless of
whether it includes one or two fronts. If the exact frontal location was known at any time, one
could judge how well the CKE envelope (or half-center) point was associated with just one front.
But considering the disagreement in climatologies (e.g., Figure 5) and the intrinsic variability of
the front, this is impossible to test. One can, however, compute the distance from the CKE half-
power point to the southern boundary (for those points that are nearest a climatological SAF
position) and the distance with the northern boundary (for those that are nearest the PF) and
compare this to the distance between the climatological positions of these fronts. Note that the
distances must be computed along the groundtracks and not simply taken as the meridional
distance at the longitude of the CKE half-power point.
The average distance between the half-power point and either northern or southern boundary
is 541 km with a standard deviation of 196 km. The average distance between the Kim and Orsi
(2014) PF and SAF along the groundtrack passes is 706 km with a standard deviation of 407 km.
We used the Kim and Orsi (2014) front positions as these data was on a regular grid which made
interpolation to the groundtrack positions easier and it was computed over the roughly the same
time span as the CKE estimates. From these statistics, we conclude the CKE envelopes should
generally only encompass either the PF or the SAF, although even if they did not, it should not
preclude one from using statistics of the CKE half-power point to deduce shifts in the fronts,
provided they are both shifting, as has been theorized.
Another method for determining frontal position is to examine the probability of jets
occurring (Chapman, 2017a) (Figure 7).  The CKE-defined mean front positions lie within the
probability envelopes, giving more confidence that the CKE measure is providing a comparable
measure of frontal position in many areas. The only location where CKE-defined fronts don't
agree well with the probability field from Chapman (2017a) is just west of the dateline, where
two points lie between levels of high jet (and hence front) probability. However, it should be
noted that Chapman finds jets in the two areas north and south of the CKE half-power points less
than 10% of the time and that the northern cluster lies on the northern edge of the enhanced CKE
envelope. Although the half-power points are slightly south of this along these two passes, this is
due to high CKE (in excess of 200 cm$^2$ s$^{-2}$) down to 58°S, where Chapman (2017a) detects few
jets. It is unclear why Chapman (2017a) detects few jets in this region of high CKE, but it should
be noted that this represents only 1% of the samples compared.
The comparison between CKE half-power points and front climatolgies is reassuring that the
method developed in Section 2 is successfully detecting regions of high energy related to jets
around fronts. Since the movement of jet positions has been used to estimate movement of the
fronts (e.g., Chapman, 2017a), a comparable calculation with positions of high CKE seems
reasonable. The majority of the estimated half-power points follows the SAF and is most likely
due to the front (and jets) moving perpendicular to the groundtracks. This method will tend to
only detect high CKE when the front is moving from northwest-to-southeast for an ascending
pass, and from southwest-to-northeast for a descending pass. This method also only works in
regions where the front is associated with highly variable jets, which does not occur at every
longitude along the front (e.g., Chapman, 2017a).

To quantify movement of the envelope of enhanced CKE, a linear trend is fit to the 8

estimations of the half-power point from 1993-2016 for each location shown in Figures 5 and 6.
Analysis of the residuals about the trend indicated they were random (lag-1 autocorrelation $< 0.1$
for all cases), so standard error was computed by scaling the formal error from the covariance
matrix determined in ordinary least squares by the standard deviation of the residuals. This was
also scaled up to account for the degrees of freedom lost by estimating the trend by sqrt($n/n_{EDOF}$),
where $n = 8$, and $n_{EDOF} = 6$. Finally, the 90% confidence interval was computed by scaling by
1.94 for 6 effective degrees of freedom assuming a normal t-distribution of the residuals.

The results indicate considerable regional variability in the change of the half-power point

over 24 years, with large uncertainty bars (Figure 8). This is due to the substantial temporal
variability in the positions, which can be seen in Figure 4, where the leading edge of the CKE
envelope varies by over 1 degree of latitude (over 100 km) between 1993-1995 and 2011-2012.
To better see significant changes outside the uncertainty (90% confidence) interval, one can
compute the signal to noise ratio (SNR = trend/uncertainty). Examining this (Figure 9), one can
see there are some regions where the half-power point has moved southward by a significant
distance over the last 24 years (13.6% of points), but there are also points where it has moved
north (9.6%). For the majority of points (76.8%), there is no statistically significant change,
meaning no movement of the front is as likely as either a southward or northward shift due to the
high variability in 3-year positions.

## 4. DISCUSSION AND CONCLUSIONS

The results from the analysis of the positions of enhanced kinetic energy suggest no overall
shift in the frontal positions across the Southern Ocean, but some large, localized movements.
The region indicative of some southward shift between 90°E and 170°E is in approximately the
same area where Kim and Orsi (2014) and Freeman and Lovenduski (2016a) also reported large
shifts, between 1992 to 2011 and 2002 and 2014, respectively. However Freeman and
Lovenduski only examined the Polar front, and Kim and Orsi (2014) only found large shifts in
the PF and the southern ACC front. They found shifts of order 50-100 km in the SAF where the
points in this study cluster, which is considerably smaller than the individual shifts we find
between 90°E and 170°E along the SAF. However, the overall average over the region between
90°E and 170°E (-29 km per decade, or -66.7 km in 23 years), is consistent with what Kim and
Orsi (2014) found.
Kim and Orsi (2014) and Freeman and Lovenduski (2016a) also found slight northward
shifts in the front positions in the southeast Pacific, between 200°E-270°E. We find some
locations in this region, where the CKE half-power points cluster around the SAF, also have a
significant northward shift. Kim and Orsi (2014) found the shift of the SAF was about 30-40 km
between 1992 and 2011. Our results suggest larger shifts in some areas; averaged over the area,
our results are 46 km per decade to the north, or 106 km from 1993-2015, which is consistent
with the average over the region computed by Freeman and Lovenduski (2016a) from sea surface
temperature data, but for the Polar Front.
Kim and Orsi (2014) suggest that the shift of the fronts in the Indian Ocean were not directly
related to shifts in winds, but instead were caused by an expansion of the Indian subtropical gyre.
They linked the shift in the southeastern Pacific to wind changes related to mainly the Southern
Annular Mode in that region (Kim and Orsi, 2014).
Overall, this study supports the recent studies by Kim and Orsi (2014), Gille (2014), Freeman
and Lovenduski (2016a), and Chapman (2017a). All find that, while the frontal positions of the
ACC are highly variable in time, there is no statistically significant shift in the fronts to the south
on average. This study utilized a novel technique to reach this conclusion, which adds to the
robustness of evidence that there has not been a shift in the frontal positions. Thus, while the
fronts may eventually shift south in a warming climate, there is no strong evidence that it is
happening at the moment.
Other studies have shown significant positive trends in the Southern Ocean that have been
connected to the warming climate. These include changes in the ocean heat content in the upper
ocean between the 1930s-1950s and 1990s (e.g., Böning et al., 2008; Gille, 2008), increases in
the heat content of deep water between the 1990s and 2005 (e.g., Purkey and Johnson, 2010),
and increases in eddy kinetic energy in the Indian and Pacific Oceans since 1993 (Hogg et al.,
2015). Observational evidence of shifts in the winds, however, indicates that while there may be
a slight southward shift in winds during the southern hemisphere summer, the overall yearly
average shift is not significant (Swart and Fyfe, 2012). Thus, the growing consensus that fronts
have not shifted to the south, on average, is consistent with observations of no significant shift in
the yearly averaged winds.

The only evidence supporting a hypothesis that ACC fronts have shifted southward since the

1990s comes from mapping the location of contours of constant dynamic topography over time
(e.g., Sokolov and Rintoul, 2009b; Kim and Orsi, 2014). As Gille (2014) argued and as we have
demonstrated based on a simple thought experiment (Figure 1), there are other equally plausible
explanations for the apparent southern shift of the contours. Considering that four different
techniques – location of mean transport (Gille, 2014), maximum SST gradients (Freeman and
Lovenduski, 2016a), probability of jet positions (Chapman, 2017a), and the location of enhanced
kinetic energy (this study) – all agree that the fronts have not moved significantly on average,
one has to conclude that the method of using dynamic topography contours to detect changes in
front position is too sensitive to sea level rise be useful for determining shifts in frontal positions,
although it may prove useful for determining the mean position as Chapman (2017a) has argued.


## Acknowledgements

The author would like to thank Christopher Chapman and an anonymous reviewer for their extensive comments on an earlier draft of this paper. Their many suggestions helped the author improve the paper substantially. This research was carried out under grant number NNX13AG98G from NASA and a grant from NOAA for the NASA/NOAA Ocean Surface Topography Science Team.

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

**Figure Captions**

**Figure 1.** a) Mean dynamic topography in the Southern Ocean along a north-south meridian for three scenarios, and b) the corresponding geostrophic velocity, with positive values indicating eastward flow. The scenarios are: an initial state (dashed black line), a shift of the two fronts south by 60 km with no change in magnitude or shape of the currents (red line), and no shift of the mean of the current, but a change in the magnitude and shape (blue line).

**Figure 2**. Positions of the T/P, Jason-1, Jason-2 and Jason-3 groundtracks used for this study (black lines), and the the approximate locations of the Subantarctic Front (red line) and the Polar Front (blue line) as estimated by Orsi et al. (1995). The orange track shows the location of the pass used in analysis shown in Figures 3 and 4.

**Figure 3.** An example profile of mean CKE (1993-2016) along a ground track in the southern Indian Ocean (shown in orange in Figure 2), demonstrating the location of the half-power point and the locations of the southern and northern boundaries of the enhanced CKE envelope. See text for details of the computations.

**Figure 4**. Three-year averages of CKE estimated along pass shown in Figure 2 (solid lines) along with the long-term mean from 1993-2016 (dotted line).

**Figure 5.** Examples of the three types of CKE profiles found (black lines), along with the value of the full EKE computed at crossover points.

**Figure 6.** Mean positions of fronts estimated from CKE (orange dots) along with estimates from other authors: Orsi et al. (1995) computed using hydrographic sections, Kim and Orsi (2014) based on contours of dynamic topography, and Freeman and Lovenduski (2016a) based on gradients of sea surface temperature. The Orsi et al. (1995) fronts were downloaded from https://gcmd.nasa.gov/records/AADC_southern_ocean_fronts.html. The Freeman and Lovenduski fronts were downloaded from https://doi.pangaea.de/10.1594/PANGAEA.855640 (Freeman and Lovenduski, 2016b). The Kim and Orsi (2014) fronts were provided by Yong Sun Kim upon request.

**Figure 7.** Mean positions of fronts estimated from CKE (black dots) along with the percent occurrence of a jet between 1993 and 2014 computed by Chapman (2017a). Data were downloaded from http://dx.doi.org/10.5061/dryad.q9k8r (Chapman, 2017b). The percent occurrence of the jet was computed by calculating the number of times a jet occurred in the daily files, dividing by the total number of days between January 1993 and December 2014, and multiplying by 100.

**Figure 8.** Estimated trend in the half-power point of CKE for each location shown in Figures 6 and 7, as a function of latitude. Error bars represent the 90% confidence interval.

**Figure 9.** SNR (trend/error in Figure 8). Values larger than 1 indicate a statistically significant northern shift. Values smaller than -1 indicate a statistically significant southern shift. Values between ± 1 indicate no statistically significant shift.

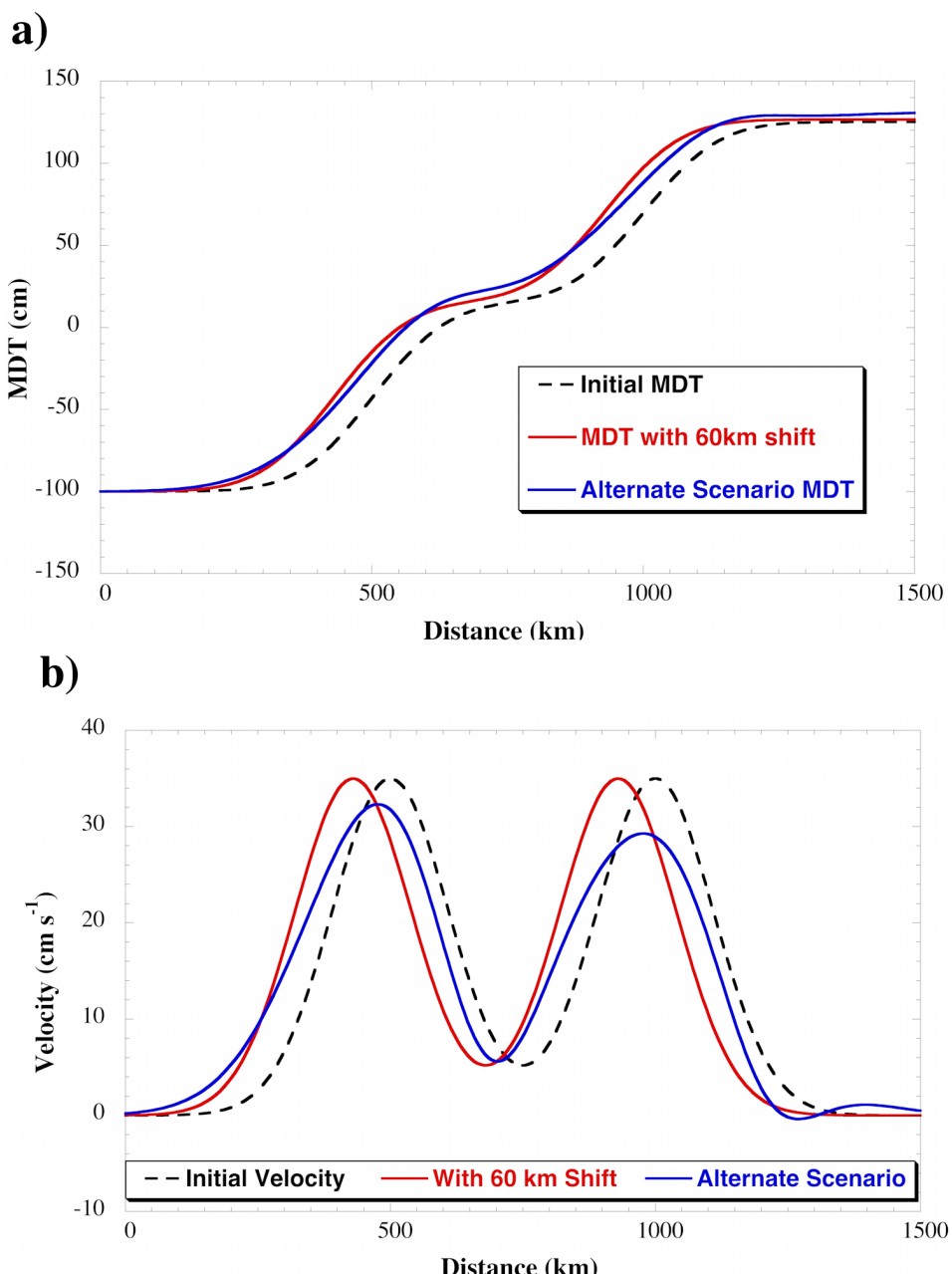

**Figure 1.** a) Mean dynamic topography in the Southern Ocean along a north-south meridian for
three scenarios, and b) the corresponding geostrophic velocity, with positive values indicating
eastward flow. The scenarios are: an initial state (dashed black line), a shift of the two fronts
south by 60 km with no change in magnitude or shape of the currents (red line), and no shift of
the mean of the current, but a change in the magnitude and shape (blue line).


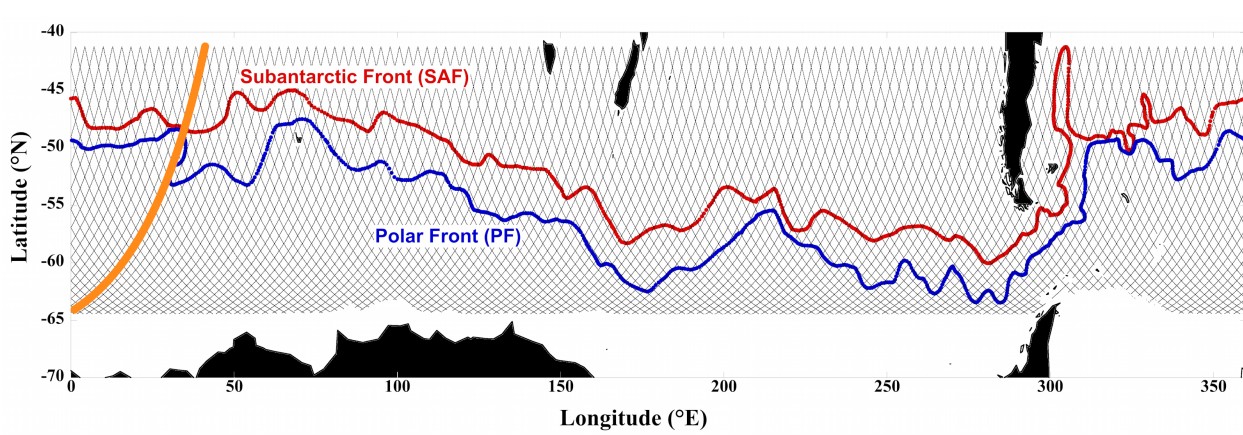

**Figure 2**. Positions of the T/P, Jason-1, Jason-2 and Jason-3 groundtracks used for this study
(black lines), and the the approximate locations of the Subantarctic Front (red line) and the Polar
Front (blue line) as estimated by Orsi et al. (1995). The orange track shows the location of the
pass used in analysis shown in Figures 3 and 4.

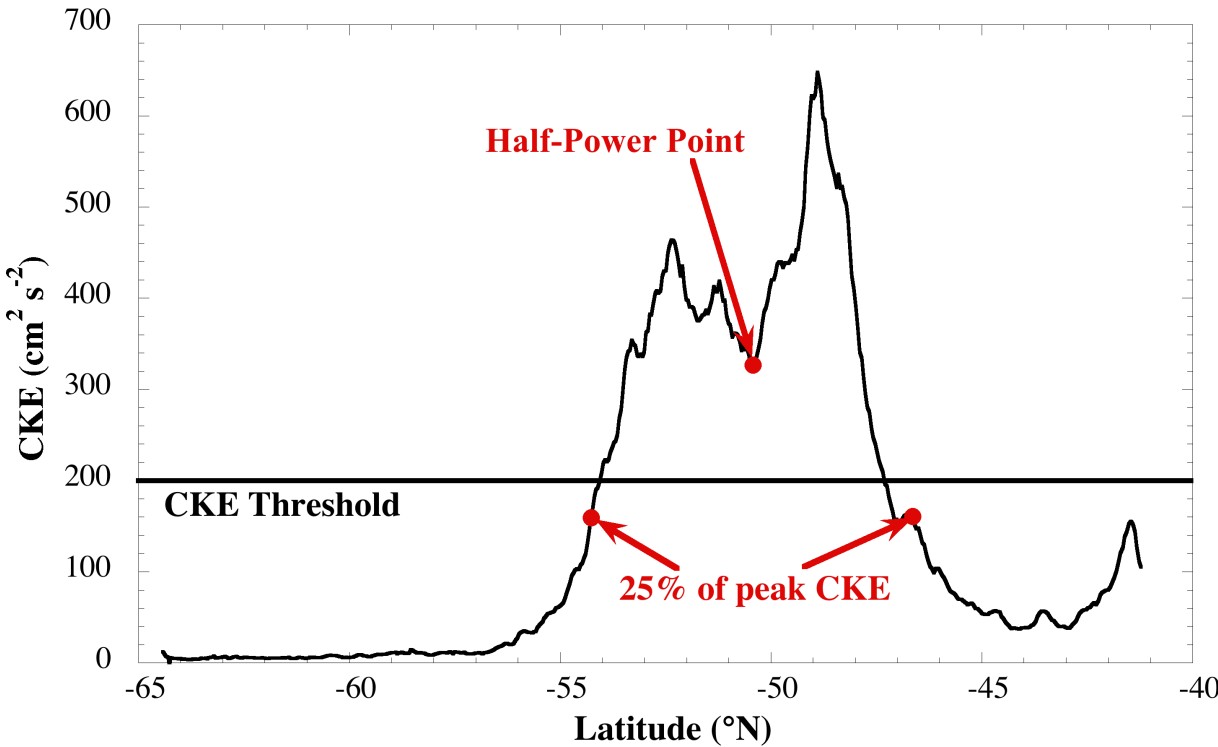

**Figure 3.** An example profile of mean CKE (1993-2016) along a ground track in the southern
Indian Ocean (shown in orange in Figure 2), demonstrating the location of the half-power point
and the locations of the southern and northern boundaries of the enhanced CKE envelope. See
text for details of the computations.


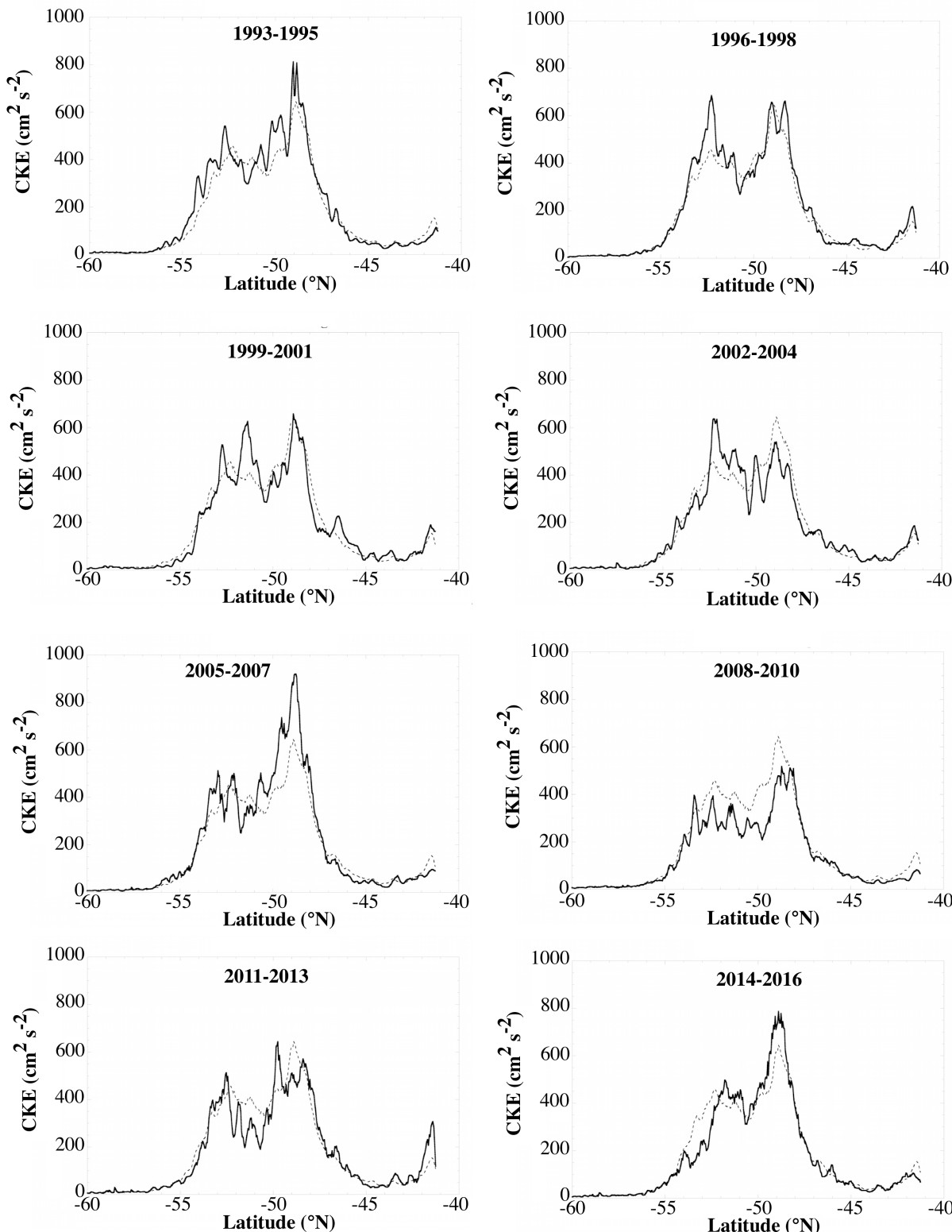

**Figure 4**. Three-year averages of CKE estimated along pass shown in Figure 2 (solid lines)
along with the long-term mean from 1993-2016 (dotted line).

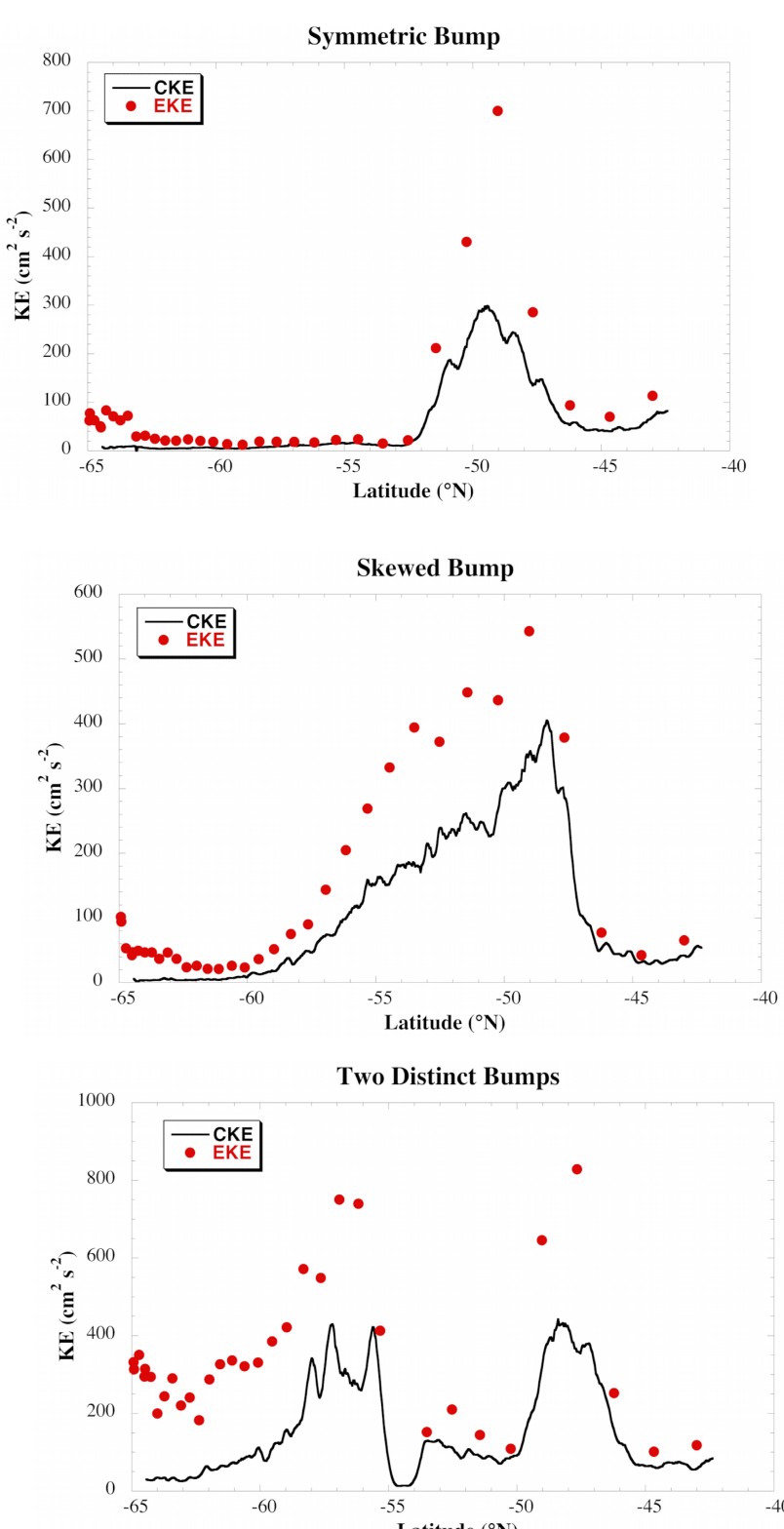

**Figure 5.** Examples of the three types of CKE profiles found (black lines), along with the value
of the full EKE computed at crossover points.


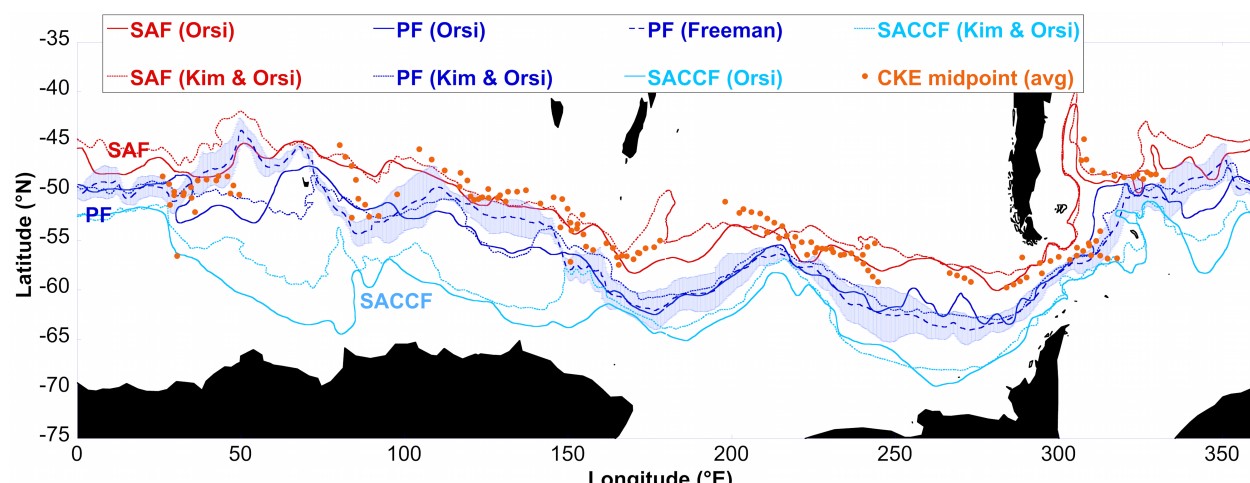



**Figure 6.** Mean positions of fronts estimated from CKE (orange dots) along with estimates from
other authors: Orsi et al. (1995) computed using hydrographic sections, Kim and Orsi (2014)
based on contours of dynamic topography, and Freeman and Lovenduski (2016a) based on
gradients of sea surface temperature. The Orsi et al. (1995) fronts were downloaded from
https://gcmd.nasa.gov/records/AADC_southern_ocean_fronts.html. The Freeman and
Lovenduski fronts were downloaded from https://doi.pangaea.de/10.1594/PANGAEA.855640
(Freeman and Lovenduski, 2016b). The Kim and Orsi (2014) fronts were provided by Yong Sun
Kim upon request.



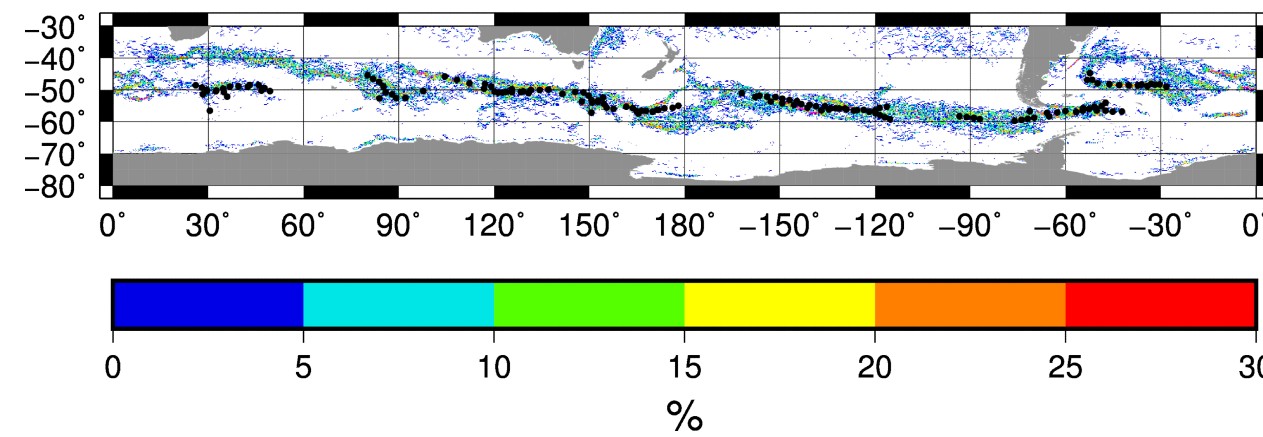

**Figure 7.** Mean positions of fronts estimated from CKE (black dots) along with the percent
occurrence of a jet between 1993 and 2014 computed by Chapman (2017a). Data were
downloaded from http://dx.doi.org/10.5061/dryad.q9k8r (Chapman, 2017b). The percent
occurrence of the jet was computed by calculating the number of times a jet occurred in the daily
files, dividing by the total number of days between January 1993 and December 2014, and
multiplying by 100.

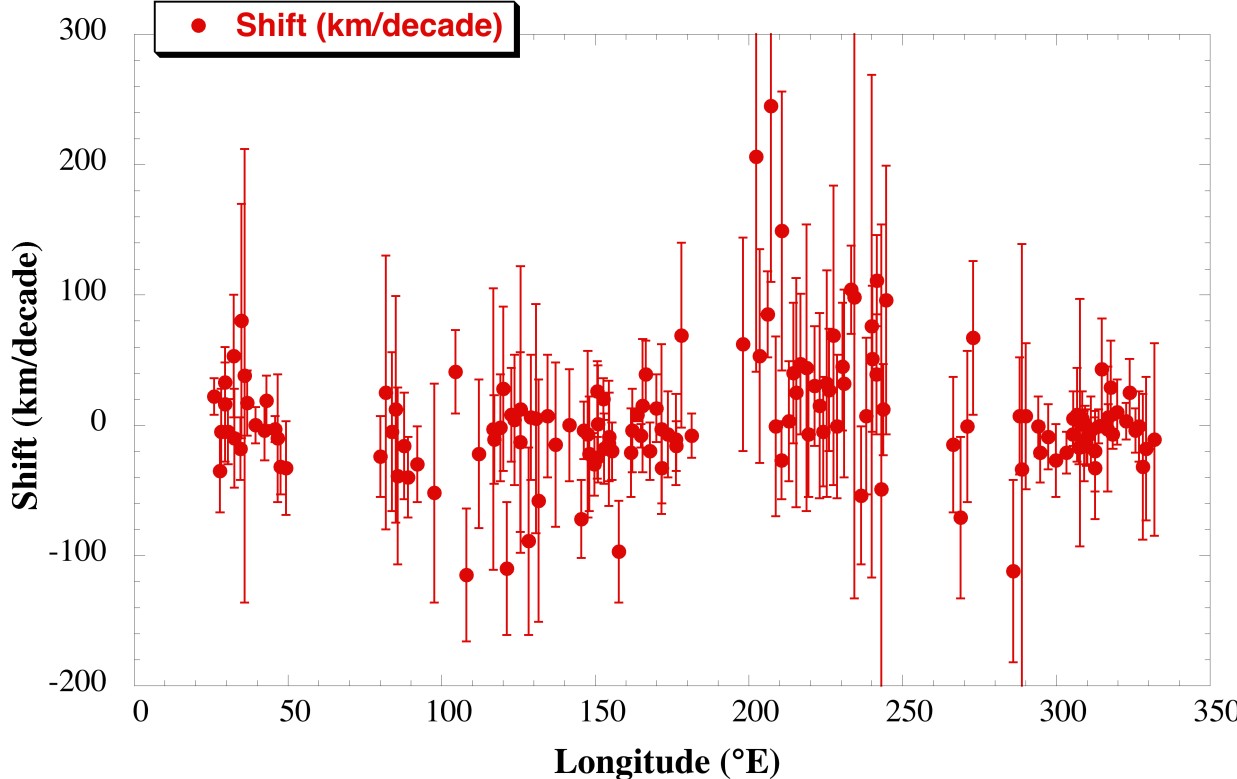

**Figure 8.** Estimated trend in the half-power point of CKE for each location shown in Figures 6 and 7, as a function of latitude. Error bars represent the 90% confidence interval.


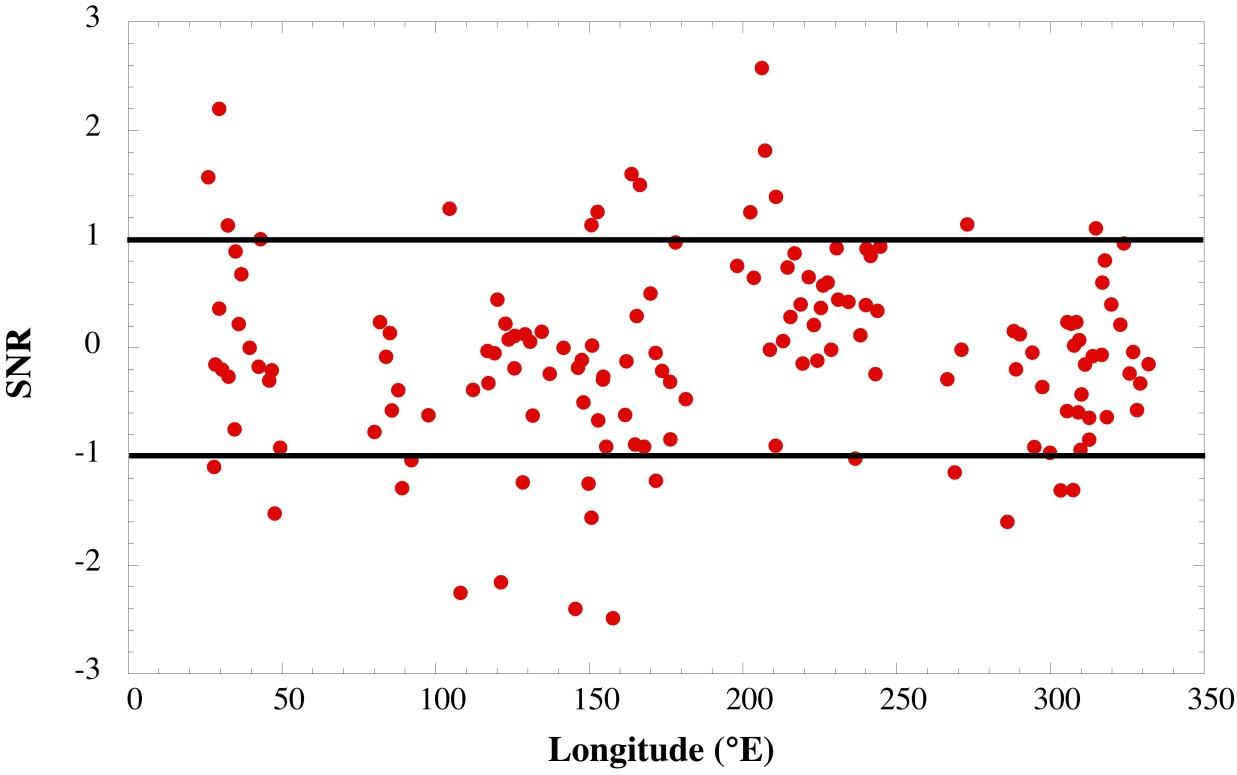

**Figure 9.** SNR (trend/error in Figure 8). Values larger than 1 indicate a statistically significant
northern shift. Values smaller than -1 indicate a statistically significant southern shift. Values
between ± 1 indicate no statistically significant shift.