# Peer review of "Using kinetic energy measurements from altimetry to detect shifts in the"

_Ocean Science, 2017_

## Referee Comment (RC1) · C. Chapman (Referee) · 29 Aug 2017

Review of Chambers 2017, Ocean Science Discussions

Summary

In this paper the author uses along track atlimetry from 1993 to 2015 to compute the geostrophic Cross Track Kinetic Energy (CKE) which is used, in turn, to infer the positions of jets and fronts in the Southern Ocean. The advantage of using the along-track altimetry, as opposed to the gridded products, is the finer spatial resolution and the fact that the optimal interpolation tends to attenuate the SSH gradients and hence the

calculated EKE. The author asserts (citing evidence from previous work) that the locations of the Southern Ocean jets can be inferred from peaks in the CKE. The author shows (convincingly) that the CKE provides a reasonable approximation of at least the form of the total kinetic energy by comparing the CKE and TKE at cross-over points.

Having established that the CKE is a reasonable approximation of the TKE, the author then uses the CKE to (a) determine the position of Subantarctic Front (SAF) and Polar Front (PF) in the Southern ocean; and (b) determine any changes in the position of the 2 fronts by repeating this calculation over 3 year periods. The author finds time-mean frontal positions that are somewhat in agreement with preceeding studies (such as the hydrographic fronts of Orsi et al. 1995) although significant differences exist in certain locations. The author also finds no significant trends in frontal locations, although there are some localised frontal shifts, which is broadly in agreement with a number of recent studies.

Overall, this study is interesting, well written and tackles a thorny problem that is currently a matter of debate. However, I have several problems with the paper as it is currently written. In particular, I am not convinced by the methodology used by the author to convert the CKE profiles into front locations, and the author either skims, or does not engage with entirely, a large amount of recent literature on this subject, and hence, an incomplete picture of the state-of-the-art is presented. My concerns will be detailed below.

The good news is that with a bit more attention to detail, and some demonstration of the methodology, this paper will become a valuable addition to the literature. I don't think this will require too much additional work.

Major comments: Engagement with prior literature

The author argues that what I will call the 'contour' view of Southern Ocean fronts (as advanced by the various papers by Sokolov & Rintoul and used in numerous studies since) gives a spurious rise to a spurious southward movement of the various fronts.
While I have no problem with this interpretation of the evidence (quite the contrary, I support it) this idea is not particularly new, and the discussion in this paper does not place the new results in the context of the extensive work that has been conducted since. While the author does cite the work of Graham et al (2012), Gille (2014) and Freeman et al. (2016), the discussion is cursoary I feel that the current state of knowledge is clearer than is presented in this paper.

For example, Thompson et al. (2010) showed using local histograms of potential vorticity in an eddy resolving model that the ACC jet/frontal structure undergoes rearrangement both spatially (generally in the lee of large topographic features) and temporaly as a result of localised mixing of the PV. The structural rearragement of the fronts complicates their intrpretation as contours of some quasi-conserved quantity. Graham et al (2012) studied in detail the response of fronts defined with contours (à la Sokolov & Rintoul) to shifts in high SSH/ADT regions, finding that the elevated grad(SSH) can shift substantially without a corresponding change in the location of the contour that is supposed to track it. Chapman (2014) showed that the temporal and spatial variability of fronts was strongly influenced by their definition.

Additionally, several relevant studies that directly study the shifts (or lack thereof) are not discussed in this study. For example, Shao et al. (2015) used a method based on higher order statistics to investigate trends in frontal position (and their response to climate modes like SAM and ENSO), finding essentially none. More recently in Chapman (2017) I used my method (described in Chapman 2014) together with a statistical model of the frontal occurrence maps to revisit this problem, once again finding no change.

As such, it seems like a consensus is starting to emerge – shifts detected with the contour method are likely spurious, and there has been minimal observed variability in the locations of the fronts and jets that make up the ACC. This paper would be more useful in the current context by explain where its results fit in, and what it does that other papers do not.

Methodology

In order to get the front location from the CKE along the satellite ground tracks, the author defines a kind of centroid, which he calls the "half-power point" (although a pedant might note that this is, actually, the half-energy point, since we're dealing with energy directly and not power), which is defined in Eqn. 2. The location of this "half-power point" is then taken to be representative of the location of the front.

I'm not convinced that this metric measures what the author says it does. For instance, the locations of jets and their associated fronts are usually take to local maxima of the SSH gradients (corresponding to the highest geostrophic velocities – and in this study, the peaks in the CKE). However, the example calculation presented in Fig. 3 shows that the half-power point is located between two peak in a local CKE minimum. One could argue (and I think the author does try to) that regardless of whether the half-power point is detecting the maxima or not, it's following the (approximate) centroid of the CKE distribution along that ground track, and thus is representative of the frontal "envelope" (for want of a better term). However, I don't see it in this example, where the two major peaks are around 10 degrees and can't really be considered part of the same feature.

An additional problem arises due to the different forms of the CKE profiles (Fig. 4). How does the calculation of the half-power point depend on the structure of the CKE profiles? I ask as I can imagine that, particularly in the case of the skewed profile (Fig. 4b) that the half-power point would be strongly biased and I'm not really sure what it would be measuring. On top of this, there's no real attempt to compare the new method with previous studies, save for the very cursory comparison in Fig. 2.

The author could clarify his calculation here by repeating the half-power point calculation on either a) some idealised profiles along the lines of those presented in Fig. 1; or (b) choosing some representative profiles and presenting them as examples in an expanded Fig. 3.

Additionally, it would help build the author's case if there was a more detailed comparison of the fronts defined in this work with other studies. This comparison needn't be too detailed, but a brief discussion would certainly help build confidence in the author's calculation.

Particular comments

Lines 49-50: "First, it assumes that the average of the position shift of the contours across all longitudes represents the shift at all longitudes"

I disagree. Sokolov & Rintoul (2009)b show numerous examples of localised shifts in the contours. The most notable is the approximately 10 degree shift of the contour associated with the PF-N as it traverses the Kerguelen Plateau (see their Fig. 12).

The problems with contour type methods are discussed in depth in Graham et al. (2012) and Gille (2014) and Chapman (2014,2017).

Line 53: While Kim & Orsi do find some migration of the fronts in the Indian Sector, it's worth noting a number of studies find no significant shifts (Graham et al. 2012, Shao et al. 2015, Chapman 2017).

Lines 75-78: "Here, we will utilize a new method to study the position of the fronts in the Southern Ocean, based on tracking the location of eddy kinetic energy (EKE) measured by altimetry. It is known from modeling studies that the front positions are associated with increased EKE, due to instabilities in the jets and interactions with bathymetry…."

I got very, very confused reading this article because of this paragraph. Here, the author states that he will use EKE to track fronts. This set off alarm bells in my head, because although the author is in general correct that EKE is higher along jets (see Hughes 1996), I can show you evidence from both models and altimetry that shows broad, high EKE regions spread over a wide range of latitudes that encompass several fronts, particularly in "storm track" regions. Thus if the author were to use high EKE to

detect fronts, I'd be skeptical and would probably need some convincing.

However, in section 2, the author appears to be using to the total kinetic energy, as he interpolate the mean MDT from the DTU10 to the ground track. Additionally, the simple examples presented in the Fig. 1 thought experiment seem to be based on the ADT and not the SLA.

Some clarification would be welcome here.

Line 118-120: Part of the author's justification for using the along track altimetry is that the gridded product attenuates the EKE due to the optimal interpolation used. So does optimally interpolating along the ground cause the same attenuation (albeit somewhat reduced)?

Line 146-148: "We initially tried tracking each of the maxima, but that quickly became complicated because sometimes the four or local maxima would become five, or even just one. This is likely due to the instability of the jets around the front."

While tracking maxima is complicated, it's not impossible to do. After all, it's been done in Thompson et al. (2010), Graham et al. (2012) and Chapman (2017) who showed a variable number of jets around the ACC. While this complication certainly could justify moving to the centroid method, I'd still like to see some stronger justification that it does pick up the frontal locations (or, at the least, their envelopes) - see major comments.

Line 158 (Eqn 2): The author calls this a centroid, but generally a centroid is defined as the first moment (or center of mass) and not the half-power point. Has the author attempted to use the standard definition? Does the half-power point have any advantages?

References Chapman, C. C., 2017 New Perspectives on Frontal Variability in the Southern Ocean, Journal of Physical Oceanography, 47, 1151-1168, doi: 10.1175/JPO-D-16-0222.1

Chapman, C. C., 2014: Southern Ocean jets and how to find them: Improving and

comparing common jet detection methods. J. Geophys. Res. Oceans, 119, 4318–4339, doi:10.1002/2014JC009810.

Hughes, C.W., 1996: The Antarctic Circumpolar Current as a Waveguide for Rossby Waves. J. Phys. Oceanogr., 26, 1375–1387, https://doi.org/10.1175/1520-0485(1996)026<1375:TACCAA>2.0.CO;2

Shao, A. E., S. T. Gille, S. Mecking, and L. Thompson, 2015: Properties of the subantarctic front and polar front from the skewness of sea level anomaly. J. Geophys. Res. Oceans, 120, 5179–5193, doi:10.1002/2015JC010723.
* * *

---

## Referee Comment (RC2) · Anonymous Referee #2 · 14 Sep 2017

Chambers, Don P. Using kinetic energy measurements from altimetry to detect shifts in the positions of fronts in the Southern Ocean. Ocean Science Discussions

**Summary**

Chambers develops a new method for the identification of the major ACC fronts, the Subantarctic Front and Antarctic Polar Front; while the ACC is comprised of additional fronts, these two fronts are known to carry the majority of ACC transport and are therefore deemed most important in this study. The ultimate goal of the presented work is to determine whether these fronts have shifted north-south in recent decades (using altimetry from 1993-2015) and place the reported results in the context of recent frontal

studies, both observational and modelling.

Given the previously documented relationship between elevated Kinetic Energy and ACC fronts and jets, this study uses Cross Track Kinetic Energy (CKE) to infer their locations in the Southern Ocean. First, the author provides justification for the use of along-track data over a gridded/mapped product to infer locations of ACC fronts and jets to ensure greater spatial resolution and lesser noise. Second, the author shows that temporally-averaged CKE-derived front locations are generally found near the traditional Orsi et al. (1995) front locations, less some gaps where the Chambers methodology is unable to locate them. Finally, the author presents a trend analysis of the CKE-estimated front locations and finds that these two fronts have not shifted south on average, in contrast to previous studies using methods that are influenced by a sea level rise signal, albeit some north-south shifts in regions of the South Indian and South Pacific, consistent with recent studies (that avoid the sea level rise issue).

**Comments**

*General*

Overall, I found the study relevant for publication in and within the scope of Ocean Science. Chambers presents novel methodology in order to independently confirm results of previous ACC studies (but offers no insight on the longevity of the method for use in future studies). While the abstract text is the most clear section of the manuscript and does a nice job of summarizing the methods and results, I take issue with the rest of the manuscript text, which lacks clarity and fluent and consistent language and voice, and in turn limits overall transparency and reproducibility of the work. Also, I'm not convinced that the author provides enough analysis of the CKE-derived front locations in order to merit its publication as is. However, with additional attention and focus on clarity and content and inclusion of a few more quantitative results, I believe this study will positively contribute to the broader literature.

*Specific*
Clarity and writing style turned out to be of significant issue. Manuscript text is jumpy and mostly written in a narrative tone rather than a scientific one (including a mixture of tenses, writing as though speaking casually, etc.) which made its reading quite difficult and confusing at times. Before publication, I recommend that text be overhauled to ensure the study is translated to the broader community effectively and to avoid losing any specific details and major findings. If text limit is not an issue, I recommend adding text and analysis where appropriate which would greatly improve the overall transparency and reproducibility of the work as well as its placement in the broader literature.

The use of 'over three-year periods' throughout the text (e.g., ln 9, lns 165-167, etc.) is inaccurate, as averages were not consistently taken over three years. As such, I would suggest that the author use 'multi-year periods,' etc. throughout the manuscript and include an explanation for those chosen. Specifically, in Figure 5, the author uses years 2011-2012 as a two-year grouping while the others before and after are three-year groupings.
(1) What is the basis for using ∼3-year periods?
(2) Does the choice of x-year groupings affect the mean positions significantly?
(3) Would this choice then affect the interpretation of the long-term trend (i.e., are the reported trends/shifts sensitive to the magnitude of groupings)?

The author motivates accurate ACC front and jet detection in light of future climate change. However, by failing to make a clear distinction between a front and a jet, the author risks adding to the already existing confusion in the literature by consistently treating them as the same thing. I feel strongly that the author should include more text on the distinction between these two physical and dynamical features, use 'fronts and jets' rather than 'fronts/jets' throughout the text (e.g., ln 31), and strive to make it clear when a front or a jet is being referenced and maintain consistency throughout. Even after completing this review, I am still unsure whether this analysis sought to detect shifts in fronts or jets, despite the title.

Moreover, while I agree this method is novel, I feel it is misleading/inaccurate to say that this study locates fronts themselves, but rather, like Gille (2014), identifies and uses a proxy for frontal and jet-like features. Please comment on this distinction. Also see specific in-line (ln 152) comment below.

Given the relatively higher resolution of the along-track data, please comment on how the presence of small-scale features (e.g., eddies) might affect the methodology and/or results, if any?

Given what we know of the influence of the depth of the ocean on ACC front and jet positioning, please comment on any quantitative assessments relating to seafloor topography? For instance, (1) are identifications of fronts and jets more successful near shallow regions or (2) is the magnitude of the 'uncertainty' in the trends shown in Figure 6 influenced by the depth of the ocean?

If possible, please comment on how this newly-developed methodology compares (in skill, accuracy, etc.) to previous front-detection methodologies and the recommendation, if any, for its future use?

Increasing the size of Figures 1 and 4 would greatly improve readability of axes.

While it is not a new idea that the methods of the previous studies tracking contours are affected by sea level rise, I like the inclusion of Figure 1 which nicely illustrates the described sensitivity.

*In-line*

lns 34-37
Please include relevant citations.

ln 71
I could not find the citation in the References section for Freeman et al. (2016).

ln 75

As the author has developed this new method, they should highlight it (e.g., 'Here, we utilize a new method…' should most definitely read 'Here, we develop a new method…')!

lns 75-79
The motivation behind using KE measurements is presented in a sloppy manner in this last paragraph of the introductory text. This motivation should be made stronger and clearer.

ln 87
I think the inclusion of the word 'high' is a typo here.

lns 83-39
Perhaps some rearranging of text is needed? The author motivates and suggests that the study uses EKE but then immediately throws it out in this section.

ln 104
Please provide the citation(s) for (and/or why) these corrections (are recommended).

lns 100, 105
Please make clearer the explanation for the interpolation method and model used. Here, it reads as if the author uses the DTU10 model to create the interpolated data (ln 100) but also that this model is then subtracted from that interpolated data (ln 105). Is it a model or model output?

ln 136
Please provide the longitude of the south Indian Ocean track used throughout the study. (If a reader were to attempt to reproduce the method, this would provide a perfect case study to check their progress.)

lns 137-143
Was there a particular optimization technique used to hone in on 200 cm$^2$ s$^{-2}$?

Further, please comment on to what extent this method may 'miss' the parts of fronts

and jets that lose energy and disappear or weaken? In other words, please comment on the limitations of this choice of threshold.

lns 146-148
Are there any more plausible explanations for the varying number of local maxima other than 'due to the instability of jets around the front' and as such, I'm not sure if I understand the author's meaning here - please explain or provide a relevant citation.

lns 147-150
Has the author performed any analyses that would serve to 'ground-truth' the assumption that the 'mean of the region of high CKE followed the front position' (i.e., using data to confirm)? Or is this purely motivated by a previous study that has already shown this but is not included as a citation?

ln 152
I'm not convinced that this method is identifying particular fronts, or at least distinguishing them from one another, as suggested (but it's possible that lack of clarity is influencing my interpretation). The author details Figure 3 as if there's only one front represented by the two peaks contained within the 'one bump' (where CKE > 200 units). However, the two peaks shown in Figure 3 could in fact be two distinct fronts, the PF (at 52S?) and the SAF (at 49S?), given the large latitudinal differences between them. Perhaps finding the mid-point in this example is really just finding the energetic space (possibly filled with weaker fronts and/or jets as suggested) in between these two major fronts. If so, this study is more like Gille (2014) than suggested (in lns 150-152): if close enough to one another, this study as presented often finds the latitude of mean CKE regardless of major front position (i.e., frontal and jet-like features, including the possibility of multiple fronts and jets of the ACC) and not the 'mean CKE around a particular front' as stated. Please comment (and elucidate the text).

Also, over what time period does Figure 3 represent? Please provide temporal averaging information.

lns 170-172
Do these calculations require the same 'simplifying assumptions' that the author refers to (and therefore avoids) earlier in the text (lns 112-114)?

ln 174
Please provide support for the author's 'reasonable assumption' conclusion.

lns 195-196
Please elaborate on or discuss the science behind the (apparent) greater number of sites of enhanced CKE found along the SAF than the PF (e.g., is the SAF known to have more KE?).

lns 197-198
What is meant by 'changes since the hydrographic data used in that study were collected?'

lns 198-199
Please provide the longitudinal location of this anomalous/southerly finding so that the reader does not have to search within the figure for it.

Here, the author presents the possibility that the method identified the SACCF to the south - please include discussion on the known high variability of the region (e.g., work by Ansorge et al., 2014)?

lns 200-206
This paragraph is the perfect opportunity to provide much-needed quantitative information. For example, in addition to referencing Figure 5 to show variability, the author could provide relevant quantities that would give the reader an idea of the 'spread' about the average across the Southern Ocean. Mean, standard deviation, etc. This information would also help to contextualize the work.

ln 201
Re: 'compared to the mean,' please provide temporal information here.

ln 203
Re: 'suggesting jets.' Why not fronts? Again, this goes back to the issue I have with the clarity of the study text. Is the author detecting fronts or jets or both with this method and if both, how are they making that distinction?

ln 205
While the author deems it 'impossible' to report on jet movements, the author could still provide the reader with some quantitative information here, such as specific comments on any temporal trends in these local maxima (e.g., their number, magnitude, etc.).

ln 210
What is meant by 'formal error?'

ln 212 Please write more mathematically. For example, instead of sqrt(8/6), '$\sqrt{n/(n-2)}$, where n is the degrees of freedom,' or the like . . . .

ln 217
Re: 'which can be seen somewhat in Figure 5,' please remove this kind of qualitative language.

ln 223
Re: 'there is no significant change,' I feel this is too strong of language. Perhaps, 'there is no statistically indistinguishable change.' The use of 'statistical' when referring to significant change is required here.

lns 227-234
No information is provided to the reader on the time periods analyzed in the referenced studies so as to make clear whether the author is making a direct comparison (also in reference to lns 239-241).

lns 235-241
The Discussion section would greatly benefit from comments on the science behind the reported/consistent northward and southward shifts over their 23-year time period.

lns 244-246

I agree. However, the clarity of the manuscript requires improvement.

lns 246-247

This is such an important statement but more content (or a rephrasing to really 'hit it home') is needed. What IS happening now, during this time of no shifts? Has there been any warming in the past 23 years? Any other changes in forcing? Please discuss more science.

ln 256

I feel the word 'flawed' is too strong here. From what I can make of it all, the studies that use the contours have results that cannot be interpreted without the caveat of sea level rise, whereas this study and the other independent studies listed do not use methods influenced by sea level rise. Therefore, instead of 'flawed' I would suggest the use of 'sensitive' to sea level rise, as Gille (2014) uses.

ln 262

Heads up: missing grant number.

**References**

Ansorge, I. J., Jackson, J. M., Reid, K., Durgadoo, J. V., Swart, S., and Eberenz, S. Evidence of a southward eddy corridor in the south-west Indian ocean. *Deep-Sea Res. Pt. II*, doi:10.1016/j.dsr2.2014.05.012, 2014.

---

## Author Comment (AC1) · 12 Oct 2017

Dear Dr. Chapman,

I appreciate your review of this paper and the obvious effort you took. Based on your comments and those of the second reviewer, I have extensively revised the paper. I have attached the fully revised paper with track changes added so you can see where I made changes.

Below I also answer your comments and describe how I have modified the paper. I'm sorry that the format that OS requires for inputting comments does not easily allow for

highlighting original comments and responses (I'm afraid I have forgotten al the LaTeX commands I ever knew), but I have tried to differentiate the original comment with REVIEWER COMMENT, my response with RESPONSE, and any additions to the text will be in quotes. Also, please note that there are some Figures with I have a attached to answer your comments.

Cheers,

Don Chambers

REVIEWER COMMENT Major comments: Engagement with prior literature The author argues that what I will call the 'contour' view of Southern Ocean fronts (as advanced by the various papers by Sokolov & Rintoul and used in numerous studies since) gives a spurious rise to a spurious southward movement of the various fronts.

While I have no problem with this interpretation of the evidence (quite the contrary, I support it) this idea is not particularly new, and the discussion in this paper does not place the new results in the context of the extensive work that has been conducted since. While the author does cite the work of Graham et al (2012), Gille (2014) and Freeman et al. (2016), the discussion is cursoary I feel that the current state of knowledge is clearer than is presented in this paper.

For example, Thompson et al. (2010) showed using local histograms of potential vorticity in an eddy resolving model that the ACC jet/frontal structure undergoes rearrangement both spatially (generally in the lee of large topographic features) and temporaly as a result of localised mixing of the PV. The structural rearragement of the fronts complicates their intrpretation as contours of some quasi-conserved quantity. Graham et al (2012) studied in detail the response of fronts defined with contours (à la Sokolov & Rintoul) to shifts in high SSH/ADT regions, finding that the elevated grad(SSH) can shift substantially without a corresponding change in the location of the contour that is supposed to track it. Chapman (2014) showed that the temporal and spatial variability of fronts was strongly influenced by their definition.

Additionally, several relevant studies that directly study the shifts (or lack thereof) are not discussed in this study. For example, Shao et al. (2015) used a method based on higher order statistics to investigate trends in frontal position (and their response to climate modes like SAM and ENSO), finding essentially none. More recently in Chapman (2017) I used my method (described in Chapman 2014) together with a statistical model of the frontal occurrence maps to revisit this problem, once again finding no change.

As such, it seems like a consensus is starting to emerge – shifts detected with the contour method are likely spurious, and there has been minimal observed variability in the locations of the fronts and jets that make up the ACC. This paper would be more useful in the current context by explain where its results fit in, and what it does that other papers do not.

Response: I have updated the Introduction extensively based on the recommendations. Please see the attached revision with track changes on. I apologize for not having seen your 2017 paper in my literature review. I have utilized the results from it in a new figure (Figure 7), which I believe should mitigate some of your concerns about this method being able to track the positions of fronts. I now show that the detected half-power points fall nicely in the probable locations of jets that you have calculated. Also, thanks for pointing out that Shao et al (2015) paper. I completely missed that one in my review.

REVIEWER COMMENT Methodology In order to get the front location from the CKE along the satellite ground tracks, the author defines a kind of centroid, which he calls the "half-power point" (although a pedant might note that this is, actually, the half-energy point, since we're dealing with energy directly and not power), which is defined in Eqn. 2. The location of this "half- power point" is then taken to be representative of the location of the front.

I'm not convinced that this metric measures what the author says it does. For instance, the locations of jets and their associated fronts are usually take to local maxima of the SSH gradients (corresponding to the highest geostrophic velocities – and in this study, the peaks in the CKE). However, the example calculation presented in Fig. 3 shows that the half-power point is located between two peak in a local CKE minimum. One could argue (and I think the author does try to) that regardless of whether the half-power point is detecting the maxima or not, it's following the (approximate) centroid of the CKE distribution along that ground track, and thus is representative of the frontal "envelope" (for want of a better term). However, I don't see it in this example, where the two major peaks are around 10 degrees and can't really be considered part of the same feature.

Response: I have included three new figures and extensive comments in the revised paper to answer your concern. First, the new Figure 4 shows the three-year averages of CKE along this pass as distinct plots to better visualize the problem of variability of the peaks. In no year is a distinct minimum CKE observed that is less than 200 cm2 s-2. Because of this and the variable number of maxima, I am hesitant to track the individual peaks, as they do not always exist.

I have added the following text in the revision, around lines 381 to 397 to justify this:

"The mean CKE profile pictured in Figure 3 has multiple local maxima, most likely associated with the variability in the narrow jets that surround the front. As shown by Chapman (2017a), these jets (evidenced in higher gradients of SSHA) do not occur around a front 100% of the time. At most, they occur about 30% of the time, and more often less than 15% of the time. Figure 4 shows the behavior of CKE along this pass for different 3-year periods. Note that the number of clearly defined maxima ranges from a low of 4 for the 2014-2016 average to 9 in 1993-1995. While other studies have estimated positions of these maxima in SSHA gradients on as short as daily intervals (e.g., Chapman, 2017a), one does not obtain a consistent number of maxima each time, making the determination of shifts difficult. Moreover, note that although there are two general peaks in CKE in the long-term mean profile, the minimum between them is still higher than 200 cm2 s-2. A minimum is also not well defined in several of the shorter averaging periods (for example, 2008-2010). Thus, instead of attempting to track all the maxima of CKE individually – analogous to tracking steepest gradients, as in Thompson et al. (2010), Graham et al. (2012), or Chapman (2017a) – we track an estimate of the center of the envelope of enhanced CKE, as it exists in all averaging periods. The assumption we make in doing this is that the localized maxima are associated with variable jets, but the position of the envelope of high CKE is related to the front."

Figures 5 and 6 in the revised paper show the estimate mean frontal positions with those from other studies, including yours (Figure 6). I feel this shows rather nicely that the positions of the half-power point of enhanced CKE falls within the estimates made by others using different techniques.

REVIEWER COMMENT An additional problem arises due to the different forms of the CKE profiles (Fig. 4). How does the calculation of the half-power point depend on the structure of the CKE profiles? I ask as I can imagine that, particularly in the case of the skewed profile (Fig. 4b) that the half-power point would be strongly biased and I'm not really sure what it would be measuring. On top of this, there's no real attempt to compare the new method with previous studies, save for the very cursory comparison in Fig. 2.

The author could clarify his calculation here by repeating the half-power point calculation on either a) some idealised profiles along the lines of those presented in Fig. 1; or (b) choosing some representative profiles and presenting them as examples in an expanded Fig. 3.

Response: Comment on skewed example, the half-power is biased toward the peak away from a simple mid-point computed from the average of the southern and northern points of the envelope. This is preferred, as it means the location is closer to the peak CKE where one would assume the strongest currents associated with the front are located.

The following text as been added to the manuscript around lines 398-428 to address this:

"There are many different ways to compute a "center" of the envelope, ranging from the average of the two end points, to a centroid calculation, to computing the point where the integral of CKE over distance is balanced on both sides, which we call the "half-power point." We have selected the latter to use, as it defines a "center" closer to the peak of CKE in the envelope. This is advantageous when the CKE curve is slightly skewed, with less magnitude on one side and more on the other. Assuming that the variability (and hence CKE) would be highest near the front (i.e., what is assumed in studies using the gradient method), finding a center of the envelope that is biased toward peak CKE is a reasonable approach." Additionally, we have included statistics on the various percentage of CKE shapes found (lines 487-494 in the revised text):

"Four general types of enhanced CKE were found (Figure 5). In most regions, the envelope in CKE is more or less symmetrical (52% of cases). Only a few profiles have two distinct regions of enhanced CKE were identified, with a clearly defined minimum below 200 cm2 s-2 between them in all time periods (3% of cases). 20% of the passes have multiple peaks that vary in time (i e., Figure 4), while 25% have a skewed envelope (Figure 5), with a long rise in CKE a long rise followed by a sharp drop-off. In all cases, though, the shape of the CKE envelope closely follows that of EKE, although the amplitude was attenuated, by anywhere from 25-50%. Having closer samples of CKE, however, allows for a better computation of the half-power point."

REVIEWER COMMENT Additionally, it would help build the author's case if there was a more detailed comparison of the fronts defined in this work with other studies. This comparison needn't be too detailed, but a brief discussion would certainly help build confidence in the author's calculation.

Response: See new Figures 6 and 7, along with the new discussion between lines 495

and 605):

"Figure 6 shows the locations of the half-power points determined from the mean CKE profiles, along with estimate of the front position based on different methods: density gradients from historical hydrographic sections (Orsi et al., 1995), dynamic topography contours (Kim and Orsi, 2014), and the gradient of sea surface temperature (Freeman and Lovenduski, 2016a). There are two estimates of the SAF and SACCF, and three of the PF. One of the PF estimates (from Freeman and Lovenduski, 2016a) includes the standard deviation of the daily estimates.

It is important to note the large differences in the estimates for the same front, which indicates how uncertain these calculations are. For instance, in the Indian Ocean at $50°E$, Freeman and Lovenduski (2016a) find the PF at the same location that Orsi et al. (1995) found the SAF, while Kim and Orsi (2014) find it significantly farther south. The SAF determination using the contour method (Kim and Orsi, 2014) is substantially farther north than the one determined from hydrographic data (Orsi et al., 1995) at most longitudes.

Many estimates from the half-power points of enhanced CKE occur between the same front estimated by different methods, indicating they are at least within the uncertainty bounds of frontal detection by any method. Other values are at locations either north or south of the other front estimates by as much as $3°$, but it should be noted that the standard deviation of the PF estimated by Freeman and Lovenduski (2016a,b) averages 2-3°, indicating these positions estimated from CKE are within the level of expected frontal variability.

Probably a better method for determining frontal position is to examine the probability of jets occurring (Chapman, 2017a) (Figure 7). The CKE-defined mean front positions lie within the probability envelopes, giving more confidence that the CKE measure is providing a comparable measure of frontal position in many areas. The only location where CKE-defined fronts don't agree well with the probability field from Chapman

(2017a) is just west of the dateline, where two points lie between levels of high jet (and hence front) probability.

Still, the good comparison is reassuring that the method developed in Section 2 is successfully detecting regions of high energy related to jets around fronts. Since the movement of jet positions has been used to estimate movement of the fronts (e.g., Chapman, 2017a), a comparable calculation with positions of high CKE seems reasonable."

REVIEWER COMMENT Lines 49-50: "First, it assumes that the average of the position shift of the contours across all longitudes represents the shift at all longitudes" I disagree. Sokolov & Rintoul (2009)b show numerous examples of localised shifts in the contours. The most notable is the approximately 10 degree shift of the contour associated with the PF-N as it traverses the Kerguelen Plateau (see their Fig. 12). The problems with contour type methods are discussed in depth in Graham et al. (2012) and Gille (2014) and Chapman (2014,2017).

Response: Agree. I have revised this section extensively. The revised text now reads (lines 69-99 of revised text):

"Using the contour method and tracking how the dynamic topography contours associated with a front position shift in time, Sokolov and Rintoul (2009b) found that the SAF and PF had both moved south by approximately 60 km over 15 years between 1993 and 2008). Kim and Orsi (2014) recently updated this analysis and found that while the average frontal position across the Southern Ocean indicates a strong southward shift, this is due to primarily to substantial shifts only in the Indian Ocean sector. They found no significant shifts throughout the Pacific or Atlantic Ocean sectors using this contour method.

The primary assumption of these analyses if a contour of dynamic topography shifts south, it is uniquely caused by front moving south. This is not true. Gille (2014) recently demonstrated that all contours in the Southern Ocean have shifted south on average, and that this follows from the observed rise in sea level – as the sea surface height rises, the contours will appear to shift south. While this breaks down at the far south and north of the ACC when dynamic topography gradients are small, these areas are far away from the PF and SAF. Gille (2014) used a different measure to determine the position of the ACC fronts, based on the latitude of the mean surface transport of the ACC measured by altimetry, which is in essence a mean location of all the jets in the Southern Ocean. She found no significant shift on average, but considerable interannual variability, especially regionally."

REVIEWER COMMENT Line 53: While Kim & Orsi do find some migration of the fronts in the Indian Sector, it's worth noting a number of studies find no significant shifts (Graham et al. 2012, Shao et al. 2015, Chapman 2017).

Response: Noted and changed in the revision. The new text is (lines 107-155 of revised text):

"Researchers using other methods also find little or no southern migration of the fronts or jets in the Southern Ocean as a whole. Graham et al. (2012) used a high-resolution model to show that the Polar Front and Subantarctic Front are constrained by bathymetry, even in increasing and shifting winds. Gille (2014) found no significant change in the latitude of mean transport of the surface currents in the ACC. Shao et al. (2015) utilized the skewness of sea level anomalies to identify front positions, and found no southward motion, but did find changes in the east Pacific correlated with the Southern Annual Mode. Chapman (2017), using positions of fronts determined from the probability of jet locations, also found no significant southward movement, but high interannual variability. Finally, Freeman et al. (2016) used weekly estimates of the Polar Front position determined from satellite sea surface temperature (SST) gradients to show no significant southward shift between 2002 and 2014 on average, except in the Indian Ocean. They also found a statistically significant northward shift of the PF in part of the south Pacific."

REVIEWER COMMENT Lines 75-78: "Here, we will utilize a new method to study the position of the fronts in the Southern Ocean, based on tracking the location of eddy kinetic energy (EKE) measured by altimetry. It is known from modeling studies that the front positions are associated with increased EKE, due to instabilities in the jets and interactions with bathymetry ... ." I got very, very confused reading this article because of this paragraph. Here, the author states that he will use EKE to track fronts. This set off alarm bells in my head, because although the author is in general correct that EKE is higher along jets (see Hughes 1996), I can show you evidence from both models and altimetry that shows broad, high EKE regions spread over a wide range of latitudes that encompass several fronts, particularly in "storm track" regions. Thus if the author were to use high EKE to detect fronts, I'd be skeptical and would probably need some convincing. However, in section 2, the author appears to be using to the total kinetic energy, as he interpolate the mean MDT from the DTU10 to the ground track. Additionally, the simple examples presented in the Fig. 1 thought experiment seem to be based on the ADT and not the SLA. Some clarification would be welcome here.

Response: I apologize for the confusion and tried to make clear I was using anomalous currents throughout Section 2, and not absolute currents. The MSS from DTU10 is used to compute sea level anomalies, as I was clear to note. I never use MDT except in the one example at the front, which is just used to motivate a way that MDT contours could shift south for other reasons than sea level change.

I have revised section 2 to put the discussion of EKE and CKE closer together to hopefully avoid this confusion, and have added the word "anomalous" more throughout to remind the reader I am only discussing anomalous currents.

While the reviewer is correct that there can be some places with high EKE over large regions, this will be at a lower limit than what I examine. It will also tend to be more episodic, and not consistent from one year to another.

I've revised the section on the calculation of the CKE envelope to discuss this more. The revised text (lines 357-371) is below:

"Several criteria were utilized to quantify where the high CKE values were considered to be associated with fronts. First, we constrained the southern boundary to be 5° south of the Orsi et al. (1995) values of the PF and the northern boundary to be 5° north of the SAF. Secondly, we used a lower-limit for CKE of 200 cm2 s-2 for detection and tested that the width of the envelope of high CKE exceeded the lower-limit for at least 100 km. The requirement that the envelope be greater than 100 km was done to reduce the impact of eddies in an otherwise quiescent region, since the diameter of eddies in the Southern Ocean is about 100 km. The CKE lower-limit was determined via iteration with different limits. For each case, the average center of the CKE envelope averaged over 24-years (based on the mean of the first and last points to exceed the lower-limit) was computed and compared visually to the Orsi et al. (1995) front positions. 200 cm2 s-2 was selected because there were a significant amount of CKE envelope centers clustered around the Orsi et al. (1995) fronts and the envelopes were found for every 10-day repeat cycle. Using a higher limit resulted in fewer detections, espscially when smaller time-averages were used. Using a lower limit, we could find more potential front positions based on CKE, but many were far from the front positions estimated by Orsi et al (1995)."

Moreover, the new Figure 7 nicely shows the positions of fronts estimated from CKE using this method align nicely with the locations found by Chapman (2017) based on the probability of a jet being present.

REVIEWER COMMENT Line 118-120: Part of the author's justification for using the along track altimetry is that the gridded product attenuates the EKE due to the optimal interpolation used. So does optimally interpolating along the ground cause the same attenuation (albeit somewhat reduced)?

Response: Optimally interpolating ANY data set can lead to attenuation of signal, but this is not something that is considered by the 99% of users who utilize the gridded AVISO products. Those products uses data from multiple passes (and altimeters) as far as 30-days away and several hundred km away (albeit weighted by a temporal/spatial covariance function) to create "1-day" grids. My OI filter utilizes data from only a few minutes and up to 200 km away along the track to optimally interpolate the data to reduce the effect of noise. As we discussed in the Hogg et al. (2015) study, the difference between the EKE from the along-track OI versus that from the gridding OI was about a 60-70% attenuation in EKE.

How to quantify the attenuation of the along-track OI filter? One way to do this is to compare OI filtered data with non-filtered data, along with an estimate of the expected noise based on the estimation from 1-Hz SSH anomaly data. Below, I show a figure of the average of OI CKE compared to CKE computed from unfiltered SSH data (Figure R1, left) for one particular pass. As can be seen, the CKE from the unfiltered data is significantly higher, but I will show this is mainly due to noise in the SSH measurement. Recall that to compute the CKE, one has to take the numerical derivative of SSH data. If the data is "noisy", the noise in the derivative will be even higher. This is then multiplied by g/f to get velocity, then squared to get CKE, increasing the noise even more. Note that since f is proportional to sin(latitude), the CKE noise should get higher as one approaches the equator. This can be seen in Figure R1 (left) as the slight slope from south to north in the CKE based on unfiltered CKE.

In a 2003 paper (Chambers D. P., J. C. Ries, and T. J. Urban, Calibration and Verification of Jason-1 Using Global Along-Track Residuals with TOPEX, Marine Geodesy, Special Issue on Jason-1 Calibration/Validation, Part 1, Vol. 26, 305-318, 2003), I estimated the RMS error in 1-sec sampling Jason-1 data by comparing to coincident TOPEX measurements and tide gauges. I estimated an error in the 1 Hz data of 3.7 cm. However, I also showed a portion of this error is correlated over distances > 100 km – due to orbit errors, dry water vapor correction errors, sea state bias correction, etc. All of these will nearly be common between adjacent 1-sec bins, so will cancel out when velocity is computed. Thus, we have to only be concerned with the uncorrelated part, which I estimated to be about 2.9 cm. Assuming this is uncorrelated between adjacent bins will increase the error in the difference to 4.0 cm, which means an error in SSH anomaly gradient of 2.8 x 10-6 radians (on average). If this is converted into CKE error (Figure R1, right) one sees this noise model nicely fits the residuals between the unfiltered and OI-filtered data. This suggests the majority of the difference in the filtered and OI filtered data is noise, although there may be some attenuation of very low-signal variability (i.e., between 56°S and 54°S), but this is not considered in this study.

If one removes the noise model from the unfiltered CKE (Figure R2), it lies on top of the CKE based on the filtered SSH anomalies, suggesting the attenuation is minimal. It is definitely much smaller than the 60-70% (on average) in the EKE computed from the gridded AVISO products. The residual differences are as likely to be from noise as any real signal.

In my opinion, a long discussion of this in the paper is not warranted. However, if the reviewer and editor feel it is necessary, I would be happy to add this discussion and the figures as an appendix to the revised manuscript. At the moment, the following has been added to the revised manuscript in the middle of Section 2 (lines 224-239), after describing where to obtain data and basic processing:

"We utilize this record rather than the gridded products based on mapping SSH from multiple altimeters (e.g., Ducet et al., 2000; Pujol et al., 2016), because the along-track data have a finer resolution in space (6.9 km along the groundtrack) and we recently demonstrated that the mapped altimetry data underestimated eddy kinetic energy (EKE) throughout the Southern Ocean compared to using along-track data by as much as 60-70% (Hogg et al., 2015). While the along-track sea level anomalies are filtered to reduce noise and thus may attenuate some signal, the filtering used (described later in this section), is less than that used for the mapped data, which uses observations from as long as 20 days and 200 km away to influence the mapped value.

[Figure]

By filtering only alongtrack data, the time differences are small (a few minutes at most), and the spatial influence is less than 100 km. Tests with unfiltered data accounting for estimated random noise in the sea level anomaly data suggests attenuation of kinetic energy is minimal with this approach and, more importantly, that the shape of the kinetic energy envelope does not significantly change."

REVIEWER COMMENT Line 146-148: "We initially tried tracking each of the maxima, but that quickly became complicated because sometimes the four or local maxima would become five, or even just one. This is likely due to the instability of the jets around the front." While tracking maxima is complicated, it's not impossible to do. After all, it's been done in Thompson et al. (2010), Graham et al. (2012) and Chapman (2017) who showed a variable number of jets around the ACC. While this complication certainly could justify moving to the centroid method, I'd still like to see some stronger justification that it does pick up the frontal locations (or, at the least, their envelopes) - see major comments

Response: I have modified this discussion to acknowledge that while others have examined the position of the steepest gradient, these positions are far more variable than looking at the centroid of the enhance CKE envelope. I have also included arguments as to why I have chosen to look at the position of the centroid over time instead of the specific jets. I have also added a new figure (new Figure 4) showing the CKE for this pass for different 3-year averages to show how the local maxima vary from period to period.

The relevant new text is (lines 381-397):

"The mean CKE profile pictured in Figure 3 has multiple local maxima, most likely associated with the narrow jets that surround the front. As shown by Chapman (2017), these jets (evidenced in higher gradients of SSHA) do not occur around a front 100% of the time. At most, they occur about 30% of the time, and more often less than 15% of the time. Figure 4 shows the behavior of CKE along this pass for different 3-year periods. Note that the number of clearly defined maxima ranges from a low of 4 for the 2014-2016 average to 9 in 1993-1995. While other studies have estimated positions of these maxima in SSHA gradients on as short as daily intervals (e.g., Chapman, 2017), by doing this one does not obtain a consistent number of maxima each time, making the determination of shifts difficult. Moreover, note that although there are two general peaks in CKE in the long-term mean profile, the minimum between them is still higher than 200 cm2 s-2. A minimum is also not well defined in several of the shorter averaging periods (for example, 2008-2010).

Thus, instead of attempting to track all the maxima of CKE individually – analogous to tracking steepest gradients, as in Thompson et al. (2010), Graham et al. (2012), or Chapman (2017) – we compute the center of the envelope of enhanced CKE and track that, as it exists in all averaging periods. The assumption we make in doing this is that the localized maxima are associated with variable jets, but the position of the envelope of high CKE is related to the front."

Finally, I include new figures showing the location of the half-power points relative to other front estimation methods (Figure 6) and relative to the probability functions of Chapman (2017) (Figure 7). I hope these demonstrate that these assumptions appear valid, in that the half-power points align with locations that other studies have detected fronts.

The figures and discussion of them has been added to Section 3 and can be seen on lines 495 and 605 of the attached revised document with track changes.

REVIEWER COMMENT Line 158 (Eqn 2): The author calls this a centroid, but generally a centroid is defined as the first moment (or center of mass) and not the half-power point. Has the author attempted to use the standard definition? Does the half-power point have any advantages?

Response: This is true. I have removed the word "centroid" in the revised text, and use "half-power point" throughout. I have computed the centroid using the standard definition and compared to the "half-power point" in several cases. For most examples, they are nearly the same (1-2 6.9 km bins away). Only in the skewed CKE example is there a major difference, and in that case the half-power point is closer to the peak of CKE. In this case, I would argue the half-power point is better, as it is closer to what one would determine using maximum gradient methods. I have added text in Section 2 to describe this argument.

The relevant text is:

"Thus, instead of attempting to track all the maxima of CKE individually – analogous to tracking steepest gradients, as in Thompson et al. (2010), Graham et al. (2012), or Chapman (2017) – we track an estimate of the center of the envelope of enhanced CKE, as it exists in all averaging periods. The assumption we make in doing this is that the localized maxima are associated with variable jets, but the position of the envelope of high CKE is related to the front.

There are many different ways to compute a "center" of the envelope, ranging from the average of the two end points, to a centroid calculation, to computing the point where the integral of CKE over distance is balanced on both sides, which we call the "half-power point." We have selected the latter to use, as it defines a "center" closer to the peak of CKE in the envelope. This is advantageous when the CKE curve is slightly skewed, with less magnitude on one side and more on the other. Assuming that the variability (and hence CKE) would be highest near the front (i.e., what is assumed in studies using the gradient method), finding a center of the envelope that is biased toward peak CKE is a reasonable approach."

Please also note the supplement to this comment:
https://www.ocean-sci-discuss.net/os-2017-57/os-2017-57-AC1-supplement.pdf
* * *
[Figure]

**Figure R1. (left)** CKE averaged from 1993-2016 along a part of a satellite pass in the Southern Ocean for SSH anomalies that have been optimally interpolated (black line) and from the raw SSH anomalies with no filtering (red line). **(right)** residuals between unfiltered CKE and CKE based on OI-filtered data, along with a noise model based on estimate uncorrelated error in Jason-1 SSH anomalies between 1-sec bins.

**Fig. 1.** (left) CKE averaged from 1993-2016 along a part of a satellite pass in the Southern Ocean for SSH anomalies that have been optimally interpolated (black line) and from the raw SSH anomalies with no fi

**Fig. 2.** CKE averaged from 1993-2016 along a part of a satellite pass in the Southern Ocean for SSH anomalies that have been optimally interpolated (black line) and from the raw SSH anomalies with no filtering

**Supplement:**

**Using kinetic energy measurements from altimetry to detect shifts in the**

**positions of fronts in the Southern Ocean**

Don P. Chambers[1]

[1] College of Marine Science, University of South Florida, St. Petersburg, FL

Correspondence to: D. Chambers (donc@usf.edu)

**Abstract.** A novel analysis is performed utilizing cross-track kinetic energy (CKE) computed from along-track sea surface height anomalies. The mid-point of enhanced kinetic energy averaged over three-year periods from 1993 to 2016 is determined across the Southern Ocean and examined to detect shifts in frontal positions, based on previous observations that kinetic energy is high around fonts in the Antarctic Circumpolar Current system due to jet instabilities. It is demonstrated that although the CKE does not represent the full eddy kinetic energy (computed from crossovers), the shape of the enhanced regions along groundtracks is the same, and CKE

has a much finer spatial sampling of 6.9 km. Results indicate no significant shift in the front positions across the Southern Ocean, on average, although there are some localized, large movements. This is consistent with other studies utilizing sea surface temperature gradients, the latitude of mean transport, and probability of jet occurrence, but is inconsistent with studies utilizing the movement of contours of dynamic topography.

Don Chambers 10/12/2017 9:38 AM

Don Chambers 9/25/2017 11:43 AM

Don Chambers 9/25/2017 11:43 AM

Don Chambers 9/25/2017 11:44 AM

Don Chambers 9/25/2017 11:44 AM

Don Chambers 9/25/2017 11:44 AM

Don Chambers 10/12/2017 9:38 AM

[revised manuscript text omitted]

Unknown

Don Chambers 10/5/2017 10:16 AM
Don Chambers 10/5/2017 10:16 AM

---

## Author Comment (AC2) · 12 Oct 2017

I appreciate your review of this paper and the obvious effort you took. Based on your comments and those of the second reviewer, I have extensively revised the paper. I have attached the fully revised paper with track changes added so you can see where I made changes.

Below I also answer your comments and describe how I have modified the paper. I'm sorry that the format that OS requires for inputting comments does not easily allow for highlighting original comments and RESPONSEs (I'm afraid I have forgotten al the LaTeX commands I ever knew), but I have tried to differentiate the original comment

with REVIEWER COMMENT, my RESPONSE with RESPONSE, and any additions to the text will be in quotes. Also, please note that there are some Figures with I have a attached to answer your comments.

Cheers,

Don Chambers

REVIEWER COMMENT: Clarity and writing style turned out to be of significant issue. Manuscript text is jumpy and mostly written in a narrative tone rather than a scientific one (including a mixture of tenses, writing as though speaking casually, etc.) which made its reading quite difficult and confusing at times. Before publication, I recommend that text be overhauled to en- sure the study is translated to the broader community effectively and to avoid losing any specific details and major findings. If text limit is not an issue, I recommend adding text and analysis where appropriate which would greatly improve the overall transparency and reproducibility of the work as well as its placement in the broader literature.

RESPONSE: Writing has been revised to keep the same tense (all present) and to add more details on processing, locations of example tracks, and statistics of data in order to make results reproducible. Please see the revised text with track changes turned on to note changes.

REVIEWER COMMENT: The use of 'over three-year periods' throughout the text (e.g., ln 9, lns 165-167, etc.) is inaccurate, as averages were not consistently taken over three years. As such, I would suggest that the author use 'multi-year periods,' etc. throughout the manuscript and include an explanation for those chosen. Specifically, in Figure 5, the author uses years 2011-2012 as a two-year grouping while the others before and after are three-year groupings. (1) What is the basis for using âĹij3-year pe- riods? (2) Does the choice of x-year groupings affect the mean positions significantly? (3) Would this choice then affect the interpretation of the long-term trend (i.e., are the reported trends/shifts sensitive to the magnitude of groupings)?

RESPONSE: This was partly a typo. It should have been 2011-2013. But the last year was only 2-years at the time of submission. 2016 data has now been processed and all results have been updated. There is no significant change. Since all groups are now three years, I will use three-year periods. The three-year average was taken to reduce influence of annual and Southern Annular Mode variations, which has been demonstrated previously. I have added text to explain the selection of three-years for the averaging period at the end of Section 2 (lines 439-477 of revised paper):

"A similar procedure was done for CKE averaged over discrete 3-year intervals, starting in January 1993 and ending in December 2016. A 3-year average was used to reduce the influence of individual eddies on determining the envelope, and to reduce interannual variations in the front position, which have been observed in other studies at some locations (e.g., Kim and Orsi, 2014; Shao et al., 2015). In particular, Kim and Orsi (2014) and Shao et al. (2015) found significant correlation with the Southern Annular Mode, which has a quasi-biennial oscillation (Hibbert et al., 2010). By averaging over three years, we found 8 distinct, statistically uncorrelated samples of CKE for each groundtrack from which to deduce shifts in the half-power point."

I do not believe it is worth the space to evaluate different averaging periods, since all are consistent now. I have tested this, however, and find that results from two-year averages are not statistically different, if one accounts for increased autocorrelation in the residuals to the trend fit.

REVIEWER COMMENT: The author motivates accurate ACC front and jet detection in light of future climate change. However, by failing to make a clear distinction between a front and a jet, the author risks adding to the already existing confusion in the literature by consistently treating them as the same thing. I feel strongly that the author should include more text on the distinction between these two physical and dynamical features, use 'fronts and jets' rather than 'fronts/jets' throughout the text (e.g., ln 31), and strive to make it clear when a front or a jet is being referenced and maintain consistency throughout. Even after completing this review, I am still unsure whether this analysis

sought to detect shifts in fronts or jets, despite the title.

Moreover, while I agree this method is novel, I feel it is misleading/inaccurate to say that this study locates fronts themselves, but rather, like Gille (2014), identifies and uses a proxy for frontal and jet-like features. Please comment on this distinction. Also see specific in-line (ln 152) comment below.

RESPONSE: Agreed. I have added several comments in the introduction to discuss the difference between fronts and jets, and the confusion that authors refer to them interchangeably. I also comment on the fact that other studies have shown one cannot actually indetify a "front" at any specific time, only in the mean sense. In the revision, I now refer "fronts and jets" or "fronts or jets" and never use "fronts/jets." The revised text is given below (lines 42-58 in the tracked changes document):

"Because of the highly variable nature of jets and the lack of clear observational detection of fronts in some areas, the literature has become muddled over the difference between a front and a jet, primarily because the "front" is rarely observed at any specific time due to the high-variability of jets (Thompson et al., 2010; Thompson and Richards, 2011; Chapman 2014; 2017). However, even in the presence of highly variable jets, methods have been developed to determine mean fronts positions in a probabilistic sense. Thompson et al. (2010) demonstrated one could define fronts in the Southern Ocean by computing probability density functions of potential vorticity in an eddy-resolving general ocean circulation model. Chapman (2014, 2017) later showed this could also be done using localized gradients in dynamic topography (i.e., high geostrophic velocity) using satellite altimeter observations, but again, only as statistical probability. This is because these areas of enhanced gradients and velocity are more reflective of jets, which strengthen and die, appear and disappear, bifurcate and join back together. Because of this, they can only be detected on average 10-15% of the time. However, Chapman (2014, 2017) has demonstrated that, at least in a mean sense, fronts defined by mean dynamic topography contours (commonly known as the "contour method") do lie within the probability distribution inferred from "gradient"

methods."

Then at the end of the Introduction (lines 160-186):

"In this paper, we develop a new method to study variability in the position of the fronts in the Southern Ocean, based on tracking the location of envelopes of kinetic energy measured by altimetry. It is known from modeling studies that the front positions are associated with increased kinetic energy, due to instabilities in the jets and interactions with bathymetry (Thompson et al., 2010; Thompson and Richards, 2011). After demonstrating that kinetic energy computed from along-track satellite altimetry forms relatively wide envelopes of enhanced energy that occur within the probability range of jets and fronts (Chapman, 2017), we track the position of this envelope from 1993 until 2016 to quantify if the envelope has shifted south by a statistically significant amount. This is based on the assumption that if the front and jets around the front has shifted south, then the envelope of high kinetic energy should also move by a comparable amount. Since kinetic energy calculation also depends on estimating gradients of sea level anomalies, this approach is similar to other gradient methods for detecting fronts or jets (e.g., Chapman, 2014; 2017; Gille, 2014; Freeman et al., 2016). It differs from these approaches, however, in that instead of determining individual gradients and tracking these over time, it looks for regions of high gradients (i.e., high energy) surround by regions of low gradient (i.e., low energy). This allows us to detect envelopes for every time-period considered, instead of only a fraction of the time, allowing for better tracking of the change over time."

Moreover, in the rest of the paper, I am clear on using "fronts" to describe a mean location of the average current or transport (as measured by CKE), while "jets" are used to describe the smaller length-scale, but highly variable CKE peaks. I feel this is consistent with other studies (e.g., Chapman, 2017).

REVIEWER COMMENT: Given the relatively higher resolution of the along-track data, please comment on how the presence of small-scale features (e.g., eddies) might affect

the methodology and/or results, if any?

RESPONSE: I have addressed this in the following text revision in Section 2 (lines 357-371):

"Several criteria were utilized to quantify where the high CKE values were considered to be associated with fronts. First, we constrained the southern boundary to be 5° south of the Orsi et al. (1995) values of the PF and the northern boundary to be 5° north of the SAF. Secondly, we used a lower-limit for CKE of 200 cm2 s-2 for detection and tested that the width of the envelope of high CKE exceeded the lower-limit for at least 100 km. The requirement that the envelope be greater than 100 km was done to reduce the impact of eddies in an otherwise quiescent region, since the diameter of eddies in the Southern Ocean is about 100 km. The CKE lower-limit was determined via iteration with different limits. For each case, the average center of the CKE envelope averaged over 24-years (based on the mean of the first and last points to exceed the lower-limit) was computed and compared visually to the Orsi et al. (1995) front positions. 200 cm2 s-2 was selected because there were a significant amount of CKE envelope centers clustered around the Orsi et al. (1995) fronts and the envelopes were found for every 10-day repeat cycle. Using a higher limit resulted in fewer detections, especially when smaller time-averages were used. Using a lower limit, we could find more potential front positions based on CKE, but many were far from the front positions estimated by Orsi et al (1995)."

REVIEWER COMMENT: Given what we know of the influence of the depth of the ocean on ACC front and jet positioning, please comment on any quantitative assessments relating to seafloor topography? For instance, (1) are identifications of fronts and jets more successful near shallow regions or (2) is the magnitude of the 'uncertainty' in the trends shown in Figure 6 influenced by the depth of the ocean?

RESPONSE: In my opinion, discussing the front and jet positions relative to bathymetry is a little off topic to this paper, and so I have chosen not to discuss it in the revised text.
[Figure]

I think it is more important to discuss the locations relative to other estimates, which I have done.

I have plotted the locations of the mean CKE half-power points along with ocean depth to show the reviewer (Figure R1, below). In general, there is no clear pattern that emerges. In some areas, high CKE is found in very deep water with no significant bathymetry changes around it (i.e., 90°W), in other places high CKE is found in deep water between shallower bathymetry, which likely leads to stronger currents and more turbulence (30-60°E). In other areas, the points follow more moderate bathymetry (3000 – 4000 m depth, 120°E-150°E). I also found no correlation between higher variance about the trend (i.e., uncertainty) and depth.

REVIEWER COMMENT: If possible, please comment on how this newly-developed methodology compares (in skill, accuracy, etc.) to previous front-detection methodologies and the recommenda- tion, if any, for its future use? Increasing the size of Figures 1 and 4 would greatly improve readability of axes.

RESPONSE: I have done this with a two new figures (Figures 6 and 7) and a revised discussion in Section 3, between lines 495 and 605:

"Figure 6 shows the locations of the half-power points determined from the mean CKE profiles, along with estimate of the front position based on different methods: density gradients from historical hydrographic sections (Orsi et al., 1995), dynamic topography contours (Kim and Orsi, 2014), and the gradient of sea surface temperature (Freeman and Lovenduski, 2016a). There are two estimates of the SAF and SACCF, and three of the PF. One of the PF estimates (from Freeman and Lovenduski, 2016a) includes the standard deviation of the daily estimates. It is important to note the large differences in the estimates for the same front, which indicates how uncertain these calculations are. For instance, in the Indian Ocean at 50°E, Freeman and Lovenduski (2016a) find the PF at the same location that Orsi et al. (1995) found the SAF, while Kim and Orsi (2014) find it significantly farther south. The SAF determination using the contour

method (Kim and Orsi, 2014) is substantially farther north than the one determined from hydrographic data (Orsi et al., 1995) at most longitudes. Many estimates from the half-power points of enhanced CKE occur between the same front estimated by different methods, indicating they are at least within the uncertainty bounds of frontal detection by any method. Other values are at locations either north or south of the other front estimates by as much as 3°, but it should be noted that the standard deviation of the PF estimated by Freeman and Lovenduski (2016a,b) averages 2-3°, indicating these positions estimated from CKE are within the level of expected frontal variability. Probably a better method for determining frontal position is to examine the probability of jets occurring (Chapman, 2017a) (Figure 7). The CKE-defined mean front positions lie within the probability envelopes, giving more confidence that the CKE measure is providing a comparable measure of frontal position in many areas. The only location where CKE-defined fronts don't agree well with the probability field from Chapman (2017a) is just west of the dateline, where two points lie between levels of high jet (and hence front) probability. Still, the good comparison is reassuring that the method developed in Section 2 is successfully detecting regions of high energy related to jets around fronts. Since the movement of jet positions has been used to estimate movement of the fronts (e.g., Chapman, 2017a), a comparable calculation with positions of high CKE seems reasonable."

REVIEWER COMMENT: lns 34-37 Please include relevant citations.

RESPONSE: This refers to the statement about the model winds. References are given to Fyfe and Saenko (2006) and Swart and Fyfe (2012). So I don't fully understand this request. The statements that follow are my observations of the figures in the paper, so don't need a reference. I have added a reference to the particular Figure in the paper that shows this:

"It should be noted, however, that the mean position of the southern hemisphere westerlies in the models lies significantly equatorward of the true position (e.g., Figure 2 in Fyfe and Saenko, 2006)."

REVIEWER COMMENT: ln 71 I could not find the citation in the References section for Freeman et al. (2016).

RESPONSE: I apologize for the oversight. It has been added.

REVIEWER COMMENT: ln 75As the author has developed this new method, they should highlight it (e.g., 'Here, we utilize a new method . . .' should most definitely read 'Here, we develop a new method. . .')!

RESPONSE: Done

REVIEWER COMMENT: lns 75-79 The motivation behind using KE measurements is presented in a sloppy manner in this last paragraph of the introductory text. This motivation should be made stronger and clearer.

RESPONSE: This has been revised on lines 164-186:

"After demonstrating that kinetic energy computed from along-track satellite altimetry forms relatively wide envelopes of enhanced energy that occur within the probability range of jets and fronts (Chapman, 2017), we track the position of this envelope from 1993 until 2016 to quantify if the envelope has shifted south by a statistically significant amount. This is based on the assumption that if the front and jets around the front has shifted south, then the envelope of high kinetic energy should also move by a comparable amount. Since kinetic energy calculation also depends on estimating gradients of sea level anomalies, this approach is similar to other gradient methods for detecting fronts or jets (e.g., Chapman, 2014; 2017; Gille, 2014; Freeman et al., 2016). It differs from these approaches, however, in that instead of determining individual gradients and tracking these over time, it looks for regions of high gradients (i.e., high energy) surround by regions of low gradient (i.e., low energy). This allows us to detect envelopes for every time-period considered, instead of only a fraction of the time, allowing for better tracking of the change over time."

REVIEWER COMMENT: ln 87 I think the inclusion of the word 'high' is a typo here.

RESPONSE: Yes. Thanks for catching that.

REVIEWER COMMENT: lns 83-39 Perhaps some rearranging of text is needed? The author motivates and suggests that the study uses EKE but then immediately throws it out in this section.

RESPONSE: Yes, I agree. I have moved the discussion of the altimetry data specifics (download and processing) to the beginning, then move the discussion of EKE and CKE after that, to keep them together and explain why EKE computed from the along-track data is not as high-resolution, so CKE is used instead. I have also moved up the equations discussing calculating EKE into Section 2 before discussing CKE.

REVIEWER COMMENT: ln 104 Please provide the citation(s) for (and/or why) these corrections (are recommended).

RESPONSE: A reference has been added

REVIEWER COMMENT: lns 100, 105 Please make clearer the explanation for the interpolation method and model used. Here, it reads as if the author uses the DTU10 model to create the interpolated data (ln 100) but also that this model is then subtracted from that interpolated data (ln 105). Is it a model or model output?

RESPONSE: I have clarified the discussion. One can either interpolate the SSH data to a mean track using the gradients of the MSS (in bilinear interpolation), or interpolate the MSS to the SSH location using bilinear interpolation. The results are the same. The model is an average of nearly 2 decades of satellite altimetry data, so not a numerical model output. The new text is (lines 215-223):

"We utilize the 1-Hz along-track SSH data from the four al-timeters and compute sea level anomalies by interpolating the DTU10 mean sea surface model (Andersen and Knudsen, 2009; http://www.space.dtu.dk/english/Research/Scientific_data_and_models/downloaddata) to the SSH location using bilinear interpolation. The DTU10 mean sea surface model

is based on SSH from multiple altimeters averaged over 17 years in a rigorous and consistent manner (Andersen and Knudsen, 2009). T/P, Jason-1, and Jason-2 data were all included. All recommended geophysical and surface corrections (e.g., water vapor, ionosphere, sea state bias, ocean tides, inverted barometer, etc) have been applied, to correct for biases introduced by atmospheric signal refraction and sea state effects (e.g., Chelton et al., 2001)."

REVIEWER COMMENT: In 136 Please provide the longitude of the south Indian Ocean track used throughout the study. (If a reader were to attempt to reproduce the method, this would provide a perfect case study to check their progress.)

RESPONSE: Done. We have identified the specific satellite pass and also highlighted the track on the revised Figure 2.

"An example of a detected high CKE envelope is shown in Figure 3, based on the average of CKE computed from T/P-Jason satellite pass 207 in the south Indian Ocean, starting at 64.3°S near the prime meridian and going to 41.2°S and 41°E longitude between 1993 and 2015."

REVIEWER COMMENT: lns 137-143 Was there a particular optimization technique used to hone in on 200 cm2s−2? Further, please comment on to what extent this method may 'miss' the parts of fronts and jets that lose energy and disappear or weaken? In other words, please comment on the limitations of this choice of threshold.

RESPONSE: The level was determined in an ad hoc procedure to find a level where all centers found CKE envelopes were in a location near Orsi et al. (1995) front positions. I also see I neglected to mention an additional criteria, that the envelope was larger than 100 km. This was done to minimize the impact of individual eddies. The revised text is (lines 357-371):

"Several criteria were utilized to quantify where the high CKE values were considered to be associated with fronts. First, we constrained the southern boundary to be 5° south

of the Orsi et al. (1995) values of the PF and the northern boundary to be 5° north of the SAF. Secondly, we used a lower-limit for CKE of 200 cm2 s-2 for detection and tested that the width of the envelope of high CKE exceeded the lower-limit for at least 100 km. The requirement that the envelope be greater than 100 km was done to reduce the impact of eddies in an otherwise quiescent region, since the diameter of eddies in the Southern Ocean is about 100 km. The CKE lower-limit was determined via iteration with different limits. For each case, the average center of the CKE envelope averaged over 24-years (based on the mean of the first and last points to exceed the lower-limit) was computed and compared visually to the Orsi et al. (1995) front positions. 200 cm2 s-2 was selected because there were a significant amount of CKE envelope centers clustered around the Orsi et al. (1995) fronts and the envelopes were found for every 10-day repeat cycle. Using a higher limit resulted in fewer detections, espscially when smaller time-averages were used. Using a lower limit, we could find more potential front positions based on CKE, but many were far from the front positions estimated by Orsi et al (1995)."

REVIEWER COMMENT: lns 146-148 Are there any more plausible explanations for the varying number of local maxima other than 'due to the instability of jets around the front' and as such, I'm not sure if I understand the author's meaning here - please explain or provide a relevant citation.

RESPONSE: I cannot think of another possibility, and others have shown the jets around the fronts are highly variable, as referenced earlier. In the revised text, I have added a new Figure (new Figure 4) showing the CKE for this pass for different 3-year averages. I have also added more discussion on this, referencing studies that have looked at these jet positions separately and making the argument why I only examine the whole envelope.

The relevant new text is between lines 381 and 397 of the revised text:

"The mean CKE profile pictured in Figure 3 has multiple local maxima, most likely

associated with the narrow jets that surround the front. As shown by Chapman (2017), these jets (evidenced in higher gradients of SSHA) do not occur around a front 100% of the time. At most, they occur about 30% of the time, and more often less than 15% of the time. Figure 4 shows the behavior of CKE along this pass for different 3-year periods. Note that the number of clearly defined maxima ranges from a low of 4 for the 2014-2016 average to 9 in 1993-1995. While other studies have estimated positions of these maxima in SSHA gradients on as short as daily intervals (e.g., Chapman, 2017), by doing this one does not obtain a consistent number of maxima each time, making the determination of shifts difficult. Moreover, note that although there are two general peaks in CKE in the long-term mean profile, the minimum between them is still higher than 200 cm2 s-2. A minimum is also not well defined in several of the shorter averaging periods (for example, 2008-2010).

Thus, instead of attempting to track all the maxima of CKE individually – analogous to tracking steepest gradients, as in Thompson et al. (2010), Graham et al. (2012), or Chapman (2017) – we compute the center of the envelope of enhanced CKE and track that, as it exists in all averaging periods. The assumption we make in doing this is that the localized maxima are associated with variable jets, but the position of the envelope of high CKE is related to the front."

REVIEWER COMMENT: lns 147-150 Has the author performed any analyses that would serve to 'ground-truth' the assumption that the 'mean of the region of high CKE followed the front position' (i.e., using data to confirm)? Or is this purely motivated by a previous study that has already shown this but is not included as a citation?

RESPONSE: I hope that the new figures showing the location of the half-power points relative to other front estimation methods (Figure 6) and relative to the probability functions of Chapman (2017) (Figure 7) demonstrate that these assumptions appear valid, in that the half-power points align with locations that other studies have detected fronts.

REVIEWER COMMENT: ln 152 I'm not convinced that this method is identifying par-

ticular fronts, or at least distinguishing them from one another, as suggested (but it's possible that lack of clarity is influencing my interpretation). The author details Figure 3 as if there's only one front represented by the two peaks contained within the 'one bump' (where CKE > 200 units). However, the two peaks shown in Figure 3 could in fact be two distinct fronts, the PF (at 52S?) and the SAF (at 49S?), given the large latitudinal differences between them. Perhaps finding the mid-point in this example is really just finding the energetic space (possibly filled with weaker fronts and/or jets as suggested) in between these two major fronts. If so, this study is more like Gille (2014) than suggested (in lns 150- 152): if close enough to one another, this study as presented often finds the latitude of mean CKE regardless of major front position (i.e., frontal and jet-like features, including the possibility of multiple fronts and jets of the ACC) and not the 'mean CKE around a particular front' as stated. Please comment (and elucidate the text).

RESPONSE: I hope the revised manuscript and new Figures 6 and 7 alleviate these concerns. As shown more clearly in the new Figure 4, only the more northerly "peak" in that CKE profile is consistent from 3-year period to 3-year period. The southerly peak is often replaced by multiple smaller peaks (i.e., 2008-2010), suggesting these are more jets than a front. I have looked at the fronts as defined by Orsi, Kim and Orsi, and Freeman and Lovenduski at this track (Figure R2, below). As you can see, the southern bump is not associated with the PF – it is the more northerly one. In fact, the Orsi front positions put the PF and SAF nearly on top of each other at this point, whereas Kim and Orsi suggest the SAF is farther north here, and find a PF position nearly identical to Freeman. I don't believe discussing the front positions for a single profile is necessary in light of the new Figures 6 and 7, and the discussion of them. But if the reviewer and editor feel this Figure is worthwhile, I can add it.

REVIEWER COMMENT: Also, over what time period does Figure 3 represent? Please provide temporal averaging information.

RESPONSE: Average of 1993-2015 for this illustrative purpose. The figure caption has

been revised to reflect this.

REVIEWER COMMENT: lns 170-172 Do these calculations require the same 'simplify-ing assumptions' that the author refers to (and therefore avoids) earlier in the text (lns 112-114)?

RESPONSE: No. I have clarified this in the revised text where the crossover and along-track velocities are discussed together, instead of separately.

REVIEWER COMMENT: ln 174 Please provide support for the author's 'reasonable assumption' conclusion.

RESPONSE: The following text has been added to answer this on lines 251 to 299:

"This formulation assumes that the velocity field has not changed significantly between the times of the groundtracks. At high latitudes, the majority of crossovers (> 78%) have a time separation of less than 3 days. At 40°, the average propagation speed of an eddy is about 3 cm s-1 [Chelton et al., 2007], meaning the eddy would have only been displaced by 8 km at most over this period. At higher latitudes, this is even less. Considering the diameter of eddies at these latitudes are of order 100 km [Chelton et al., 2007], the movement is not large enough to cause a significant change in velocity at the point."

REVIEWER COMMENT: lns 195-196 Please elaborate on or discuss the science be-hind the (apparent) greater number of sites of enhanced CKE found along the SAF than the PF (e.g., is the SAF known to have more KE?).

RESPONSE: I don't really have a good explanation for this, and choose not to specu-late for the reason. These are the regions with enhanced CKE as found by the relatively conservative criteria we use. If I use lower CKE limits, I can find more points along the PF, but I also find many more between the PF and SACCF. Thus, I try to be conserva-tive in the limits used in the algorithm.

REVIEWER COMMENT: lns 197-198 What is meant by 'changes since the hydrographic data used in that study were collected?'

RESPONSE: That phrase has been deleted as the discussion (lines 495-595) now focuses on the wide spread among different estimates.

REVIEWER COMMENT: lns 198-199 Please provide the longitudinal location of this anomalous/southerly finding so that the reader does not have to search within the figure for it

RESPONSE: All locations of specific deviations discussed in the paper have now been added.

REVIEWER COMMENT: Here, the author presents the possibility that the method identified the SACCF to the south - please include discussion on the known high variability of the region (e.g., work by Ansorge et al., 2014)?

RESPONSE: I really don't think it is relevant to this discussion to cite that paper, considering there is only one point that might be in the SACCF. The revised paper does not explicitly highlight this point.

REVIEWER COMMENT: lns 200-206 This paragraph is the perfect opportunity to provide much-needed quantitative information. For example, in addition to referencing REVIEWER COMMENT: Figure 5 to show variability, the author could provide relevant quantities that would give the reader an idea of the 'spread' about the average across the Southern Ocean. Mean, standard deviation, etc. This information would also help to contextualize the work

RESPONSE: This section has been extensively revised, with a new figure showing the variability (Figure 4) for each 3-year period. The discussion is on lines 381-392.

"The mean CKE profile pictured in Figure 3 has multiple local maxima, most likely associated with the variability in the narrow jets that surround the front. As shown by Chapman (2017a), these jets (evidenced in higher gradients of SSHA) do not occur around a front 100% of the time. At most, they occur about 30% of the time, and more

often less than 15% of the time. Figure 4 shows the behavior of CKE along this pass for different 3-year periods. Note that the number of clearly defined maxima ranges from a low of 4 for the 2014-2016 average to 9 in 1993-1995. While other studies have estimated positions of these maxima in SSHA gradients on as short as daily intervals (e.g., Chapman, 2017a), one does not obtain a consistent number of maxima each time, making the determination of shifts difficult. Moreover, note that although there are two general peaks in CKE in the long-term mean profile, the minimum between them is still higher than 200 cm2 s-2. A minimum is also not well defined in several of the shorter averaging periods (for example, 2008-2010)."

REVIEWER COMMENT: ln 201 Re: 'compared to the mean,' please provide temporal information here.

RESPONSE: This information has been added to the figure caption.

REVIEWER COMMENT: ln 203 Re: 'suggesting jets.' Why not fronts? Again, this goes back to the issue I have with the clarity of the study text. Is the author detecting fronts or jets or both with this method and if both, how are they making that distinction?

RESPONSE: I have revised this section and added new text to clarify my argument that we are detecting fronts as defined by the envelope of enhanced CKE driven by variable jets that surround the fronts (lines 393-397):

"Thus, instead of attempting to track all the maxima of CKE individually – analogous to tracking steepest gradients, as in Thompson et al. (2010), Graham et al. (2012), or Chapman (2017a) – we track an estimate of the center of the envelope of enhanced CKE, as it exists in all averaging periods. The assumption we make in doing this is that the localized maxima are associated with variable jets, but the position of the envelope of high CKE is related to the front."

REVIEWER COMMENT: ln 205 While the author deems it 'impossible' to report on jet movements, the author could still provide the reader with some quantitative information

here, such as specific comments on any temporal trends in these local maxima (e.g., their number, magnitude, etc.).

This has been done in the revised text lines 381-392.

"The mean CKE profile pictured in Figure 3 has multiple local maxima, most likely associated with the variability in the narrow jets that surround the front. As shown by Chapman (2017a), these jets (evidenced in higher gradients of SSHA) do not occur around a front 100% of the time. At most, they occur about 30% of the time, and more often less than 15% of the time. Figure 4 shows the behavior of CKE along this pass for different 3-year periods. Note that the number of clearly defined maxima ranges from a low of 4 for the 2014-2016 average to 9 in 1993-1995. While other studies have estimated positions of these maxima in SSHA gradients on as short as daily intervals (e.g., Chapman, 2017a), one does not obtain a consistent number of maxima each time, making the determination of shifts difficult. Moreover, note that although there are two general peaks in CKE in the long-term mean profile, the minimum between them is still higher than 200 cm2 s-2. A minimum is also not well defined in several of the shorter averaging periods (for example, 2008-2010)."

REVIEWER COMMENT: ln 210 What is meant by 'formal error?'

RESPONSE: Formal error is the error that comes out of the covariance matrix of ordinary least squares when it has not been scaled by the variance of the residuals. In the ordinary computation, this assumes the variance has been normalized to 1, so does not represent the true variance of the residuals. Hence one should scale this by the variance of the residuals to the fit (at a minimum) before estimating the standard error.

Since this is a standard definition, I don't feel any more detail or references are required in the text.

REVIEWER COMMENT: ln 212 Please write more mathematically. For example, instead of sqrt(8/6),'$\sqrt{n/(n-2)}$, where n is the degrees of freedom,' or the like . . . .

RESPONSE: This has been changed to:

"This was also scaled up to account for the degrees of freedom lost by estimating the trend by sqrt(n/nEDOF), where n = 8, and nEDOF = 6."

REVIEWER COMMENT: ln 217 Re: 'which can be seen somewhat in Figure 5,' please remove this kind of qualitative language.

RESPONSE: "Somewhat" has been removed, as the new figure (Figure 4), shows this more clearly.

REVIEWER COMMENT: ln 223 Re: 'there is no significant change,' I feel this is too strong of language. Perhaps, 'there is no statistically indistinguishable change.' The use of 'statistical' when referring to significant change is required here.

RESPONSE: this has been changed to:

"For the majority of points (76.8%), there is no statistically significant change – no movement of the front is as likely as either a southward or northward shift due to the high variability in 3-year positions."

REVIEWER COMMENT: lns 227-234 No information is provided to the reader on the time periods analyzed in the referenced studies so as to make clear whether the author is making a direct comparison (also in reference to lns 239-241).

RESPONSE: The time periods have been added.

REVIEWER COMMENT: lns 235-241 The Discussion section would greatly benefit from comments on the science behind the reported/consistent northward and southward shifts over their 23-year time period.

RESPONSE: I have added a comment on this, which has been addressed in the Kim and Orsi (2014) study:

"Kim and Orsi (2014) suggest that the shift of the fronts in the Indian Ocean were

not directly related to shifts in winds, but instead were caused by an expansion of the Indian subtropical gyre. They linked the shift in the southeastern Pacific to wind changes related to mainly the Southern Annular Mode in that region (Kim and Orsi, 2014)."

REVIEWER COMMENT: lns 244-246 I agree. However, the clarity of the manuscript requires improvement.

RESPONSE: I hope the substantially revised manuscript is clearer now.

REVIEWER COMMENT: lns 246-247 This is such an important statement but more content (or a rephrasing to really 'hit it home') is needed. What IS happening now, during this time of no shifts? Has there been any warming in the past 23 years? Any other changes in forcing? Please discuss more science.

RESPONSE: I have added a paragraph on what changes have been observed in the Southern Ocean in the last two decades, and I have also rearranged some of the text in those lines. The revised and new text is:

"Overall, this study supports the recent studies by Kim and Orsi (2014), Gille (2014), Freeman and Lovenduski (2016a), and Chapman (2017). All find that, while the frontal positions of the ACC are highly variable in time, there is no statistically significant shift in the fronts to the south on average. This study utilized a novel technique to reach this conclusion, which adds to the robustness of evidence that there has not been a shift in the frontal positions. Thus, while the fronts may eventually shift south in a warming climate, there is no strong evidence that it is happening at the moment.

Other studies have shown significant positive trends in the Southern Ocean that have been connected to the warming climate. These include changes in the ocean heat content in the upper ocean since the between the 1930s-1950s and 1990s (e.g., Böning et al., 2008; Gille, 2008), increases in the heat content of deep water between the 1990s and 2005 (Purkey and Johnson, 2010), and increases in eddy kinetic energy in

the Indian and Pacific Oceans since 1993 (Hogg et al., 2015). Observational evidence of shifts in the winds, however, indicates that while there may be a slight southward shift in winds during the southern hemisphere summer, the overall yearly average shift is not significant (Swart and Fyfe, 2012). Thus, the growing consensus that fronts have not shifted to the south, on average, is consistent with no significant shift in the yearly averaged winds."

REVIEWER COMMENT: In 256 I feel the word 'flawed' is too strong here. From what I can make of it all, the studies that use the contours have results that cannot be interpreted without the caveat of sea level rise, whereas this study and the other independent studies listed do not use methods influenced by sea level rise. Therefore, instead of 'flawed' I would suggest the use of 'sensitive' to sea level rise, as Gille (2014) uses.

RESPONSE: the text has been changed to read:

"...one has to conclude that the method of using dynamic topography contours to detect changes in front position is too sensitive to sea level rise be useful for determining shifts in frontal positions, although it may prove useful for determining the mean position as Chapman (2017a) has argued."

REVIEWER COMMENT: In 262 Heads up: missing grant number.

RESPONSE: Thank you. I left that out because at the time of submission, NOAA had not established the new grant number for this research. NOAA has established the funding, but as a subaward to a larger award handled through the University of Miami, so I have just revised to "a grant from NOAA".

Please also note the supplement to this comment:
https://www.ocean-sci-discuss.net/os-2017-57/os-2017-57-AC2-supplement.pdf

[Figure]

[Figure]

**Fig. 1.** Positions of mean CKE half-power points along with ocean depth (in km). Bathymetry data from ETOPO5C.

**Average (1993-2015)**

**PF (Kim & Orsi; Freeman)**

**SAF (Kim & Orsi)**

**PF
(Orsi)**

**SAF
(Orsi)**

$CKE\ (cm^2\ s^{-2})$

Latitude (°N)

**Fig. 2.** CKE along pass 207 (same as Figure 3 in revised paper), but with front positions estimated by different groups.

---

## Author Response (AR2)

Matthew Hecht
Editor, *Ocean Science*

Dear Dr. Hecht,

Thank you for giving me a chance to respond to the further criticisms of Reviewer # 2 and to submit a suitably revised paper. After reading the review, I feel the comments derive from a misunderstanding of the intent of this paper. The method I propose is NOT intended to isolate the very narrow fronts of the Southern Ocean. It is only intended to track CHANGES in their position. This is a subtle, but important distinction. I admit that in the original draft, this was not entirely clear. But I have made every effort in the second draft to make this clear. For instance, in the Introduction, I write:

"In this paper, we develop a new method to study linear shifts in the position of the fronts in the Southern Ocean, based on tracking the location of envelopes of kinetic energy measured by satellite altimetry. It is known from modeling studies that the front positions are associated with increased kinetic energy, due to instabilities in the jets and interactions with bathymetry (Thompson et al., 2010; Thompson and Richards, 2011). After demonstrating that kinetic energy computed from along-track satellite altimetry forms relatively wide envelopes of enhanced energy that occur within the probability range of jets and fronts (e.g., Chapman, 2017a), we track the positions of these envelopes from 1993 until 2016 to quantify if the envelopes have shifted south by a statistically significant amount. This is based on the assumption that if the front and jets around the front have shifted south, then the envelope of high kinetic energy should also move by a comparable amount. We do not purport that our method derives the actual position of either a front or a jet due to the relatively wide swath of enhanced kinetic energy on either side of fronts related to variability of jets. Instead, we only purport that it can indicate shifts in the frontal position, because if a front has shifted south by 100 km (for instance), then the band of enhanced kinetic energy should also shift south by a comparable amount. It is difficult to reconcile a frontal shift without a displacement of kinetic energy."

Note that the last three sentences are new, to explicitly state that I do not intend this method to be used to determine specific frontal positions at any time. I hope this will alleviate further confusion by reviewer and reader. I also now reiterate this point in the conclusion section.

I hope you find this new revised manuscript acceptable.

Cheers,

Don Chambers

**Response to Reviewer # 1 (Christopher Chapman)**

     I appreciate your kind comments on the revised manuscript. I am glad I could answer all your concerns. Your first review was tremendously helpful.  I apologize for the quality of the figures in the PDF, but that is outside of my control as they are apparently reduced in resolution in the PDF creation by the *Ocean Science* program in order to save space. The original figure files are all high-resolution (> 500 dpi) in all cases.

**Response to Reviewer # 2**

I appreciate your review of this paper and the obvious effort you took. Based on your additional comments, I have revised the paper slightly, as indicated below in answers to your specific comments and in the attached document with track changes turned on. I have attached the fully revised paper with track changes added so you can see where I made changes.

The major criticism appears to derive from a misunderstanding of the intent of this paper. I agree with you that the method I propose should NOT be used to isolate the very narrow fronts of the Southern Ocean. It is only intended to track CHANGES in their position. This is a subtle, but important distinction. I admit that in the original draft, this was not entirely clear. But I have made every effort in the second draft to make this clear. For instance, in the Introduction, I write:

"In this paper, we develop a new method to study linear shifts in the position of the fronts in the Southern Ocean, based on tracking the location of envelopes of kinetic energy measured by satellite altimetry. It is known from modeling studies that the front positions are associated with increased kinetic energy, due to instabilities in the jets and interactions with bathymetry (Thompson et al., 2010; Thompson and Richards, 2011). After demonstrating that kinetic energy computed from along-track satellite altimetry forms relatively wide envelopes of enhanced energy that occur within the probability range of jets and fronts (e.g., Chapman, 2017a), we track the positions of these envelopes from 1993 until 2016 to quantify if the envelopes have shifted south by a statistically significant amount. This is based on previous evidence (e.g., Thompson et al., 2010; Thompson and Richards, 2011; Chapman, 2017) that kinetic energy is highest around fronts in the Southern Ocean. Thus, if the fronts have shifted south, then it follows that the envelope of high kinetic energy should also move by a comparable amount. We do not purport that our method derives the actual position of either a front or a jet due to the relatively wide swath of enhanced kinetic energy on either side of fronts related to variability of jets. Instead, we only purport that it can indicate shifts in the frontal position, because if a front has shifted south by 100 km (for instance), then the band of enhanced kinetic energy should also shift south by a comparable amount. It is difficult to reconcile a frontal shift without a displacement of kinetic energy."

Note that the last four sentences are new (or revised), to explicitly state that I do not intend this method to be used to determine specific frontal positions at any time. I hope this will alleviate any further confusion by the reviewer and the reader. I also reiterate this point in the conclusion section.

The analysis comparing the half-power points of CKE to previously estimated frontal positions is still pertinent, however, to demonstrate that the CKE regions are close to estimated frontal positions. If they were not, then it would raise questions whether this method could be used to detect shifts.

I hope these revisions, and others discussed below in response to specific criticisms alleviates any further concerns.

Cheers,

Don Chambers

*Reviewer Comment: Figures*
*1. All figure axes (except Figure 3) are still too small for clear readability.*

**RESPONSE**: I apologize for the quality of the figures in the PDF, but that is outside of my control as they are reduced in resolution in the PDF creation by the *Ocean Science* program in order to save space. The original figure files are all high-resolution (> 500 dpi) in all cases. However, some figures, particularly Figures 4, 5, and 6 do have smaller fonts because of down-scaling of the figures. I have made all of these larger. I have also made the dots in Figure 7 larger.

*Reviewer Comment: 2. Thank you for updating the analysis to include most recent 2016 data. Heads up: now there are differing time periods 1993-2015 and 1993-2016 between Figures 2-4.*

**RESPONSE**: Sorry. I missed changing the caption on Figure 2, even though I changed the figure. There was no visually apparent difference between the two long averages.

*Reviewer Comment: Lines 184-186*
*The author states, 'we develop a new method to study variability in the position of the fronts in the Southern Ocean' but no quantitative results on front variability, other than N-S shifts, are presented. Percentages are provided in lines 488-539 but this pertains to the shapes of the enhanced CKE envelopes and not the front location(s) found within the envelopes. For example, what is the standard deviation in the 8 distinct CKE half-power point values ('front latitudes') at each longitude (i.e., variability in the 'front' itself)?*

**RESPONSE**: First, we have changed the sentence to read: "we develop a new method to study linear shifts in the position of the fronts in the Southern Ocean" to emphasize we are focusing only on shifts, not other variability. Second, we have added pertinent statistics of CKE envelope width in the results section.

*Reviewer Comment: Lines 190-191*
*This is the most accurate statement of the study: 'we track the positions of these envelopes from 1993 until 2016 to quantify if the envelopes have shifted…'. Given that these envelopes can span more than 5 degrees latitude N-S, and that multiple (traditional) fronts can exist within one envelope, I take away that this study tracks shifts in a mean ACC position rather than individual fronts themselves, as claimed. The fact that the community is accustomed to thinking of fronts in a particular way, it is important that the author makes their definition of a 'front' as clear as possible.*

**RESPONSE**: We stated in the revised document immediately after that statement: "This is based on the assumption that if the front and jets around the front have shifted south, then the envelope of high kinetic energy should also move by a comparable amount."

We have revised this sentence to now read: "Since kinetic energy is highest around fronts in the Southern Ocean  (e.g., Thompson et al., 2010; Thompson and Richards, 2011; Chapman, 2017) , it follows that if the fronts have shifted south, then the envelope of high kinetic energy should also move by a comparable amount."

Although the first part of this revised sentence is essentially redundant with a statement two sentences earlier, we add it here to make it absolutely clear what is the basis for our analysis. I note that this assumption of the CKE shifting south is essentially the one made by proponents of the contour method (i.e., identify a front based on a contour that passes through a large dynamic topography gradient in one region, then track its shift). However, because CKE is based on gradients of SSH, it is insensitive to changes in large-scale sea level changes, unlike the contour method.

Finally, to make it clear we do not intend that this method be used to isolate specific fronts at any time, we now state:

"We do not purport that our method derives the actual position of either a front or a jet due to the relatively wide swath of enhanced kinetic energy on either side of fronts related to variability of jets. Instead, we only purport that it can indicate shifts in the frontal position, because if a front has shifted south by 100 km (for instance), then the band of enhanced kinetic energy should also shift south by a comparable amount. It is difficult to reconcile a frontal shift without a displacement of kinetic energy."

*Reviewer Comment: Line 214*
*Describe these 'envelopes' more. I appreciate Figures 3 and 4 but this is at one specific pass in the Indian sector. How do these envelopes vary by region, etc.? What is the average meridional width/extent, etc.? What is the standard deviation in the width of the envelope at each longitude?*

**RESPONSE**: We have added some discussion and statistics in response to this one and similar comments in Section 3 (Results and Analysis). We feel it is more relevant in this section than in the methods section. The main issue the reviewer has with the analysis appears to be concern that the width of the CKE envelope is considerably wider than the width of a single front, or that the shift in the CKE half-power point is narrower than the width of the envelope. The latter really should not be a concern. Tracking means of a distribution is a commonly utilized and robust statistical test, and we allow for appropriate uncertainty estimates that are described at the end of the Section 3. Even if an envelope is quite large, the mean (i.e., half-power point in this case) can be a robust and significant measure of its location. The envelope will definitely be wider than the width of the front, but we have addressed this in numerous previous comments and revisions, noting the envelope will exist around the fronts and not isolate a specific location of the front.

Another concern the reviewer appears to have is that the width may be greater than the distance between fronts, thus the CKE envelope may contain both fronts. This is indeed a possibility, but as we have argued, without a definitive location of a front (and climatologies differ considerably on locations), we can only use an objective measure of the approximate front location. Here, we use the envelope of enhanced CKE, without distinguishing whether this envelope is about the SAF or PF. Our argument is simply that if the fronts have moved south (which is claimed by some), then this envelope MUST also move south.

One can compare the width to the distance between fronts, and we now supply this statistics and some discussion in Section 3. However, it is not the full-width of the CKE envelope one should compare to distances between fronts, but the half-width, assuming the half-power point is in the middle of the envelope, and the "front" is also located somewhere within the envelope. Moreover, one can't compare this to the meridional distance between fronts, since the altimeter is sampling the ocean along an inclined groundtrack, leading to a longer arc of distance. One needs to compute the distance between fronts along the groundtrack. We did this, based on the Kim and Orsi climatology, since it is based on a regular gridding.

We found that the average half width of the CKE envelope was 541 km (standard deviation = 196 km), whereas the average distance between the SAF and PF along each groundtrack was 706 km, with a standard deviation of 407 km. Thus, the mean half-width of the CKE envelope was less than the mean distance between the PF and SAF, although the latter distance is more variable.

I have added two paragraphs to the Discussion in Section 3 on this:

"One may question whether the relatively wide envelopes of enhanced CKE overlap more than one front. This is a possibility, but if both fronts have moved south as some have argued (e.g., Sokolov and Rintoul, 2009b), then the CKE envelope should also shift, regardless of whether it includes one or two fronts. If the exact frontal location was known at any time, one could judge how well the CKE envelope (or half-center) point was associated with just one front. But considering the disagreement in climatologies (e.g., Figure 5) and the intrinsic variability of the front positions, this is impossible to test. One can, however, compute the distance from the CKE half-power point to the southern boundary (for those points that are nearest a climatological SAF position) and the distance with the northern boundary (for those that are nearest the PF) and compare this to the distance between the climatological positions of these fronts. Note that the distances must be computed along the groundtracks and not simply taken as the meridional distance at the longitude of the CKE half-power point.

The average distance between the half-power point and either northern or southern boundary is 541 km with a standard deviation of 196 km. The average distance between the Kim and Orsi (2014) PF and SAF along the groundtrack passes is 706 km with a standard deviation of 407 km. We used the Kim and Orsi (2014) front positions as these data were on a regular grid which made interpolation to the groundtrack positions easier and it was computed over the roughly the same time span as the CKE estimates. From these statistics, we conclude the CKE envelopes should generally only encompass either the PF or the SAF, although even if they did not, it should not preclude one from using statistics of the CKE half-power point to deduce shifts in the fronts, provided they are both shifting, as has been theorized."

*Reviewer Comment: Lines 348-369*
*Please comment on the reasoning behind the choice to bias the methodology to the Orsi study, in particular.*

**RESPONSE**: We do not believe that the method of initial detection "biases" the results to the Orsi front positions, as indicated by Figure 6. If we had used any other estimate, we would have found the same locations. We have added a statement at the end of the paragraph to make this clear:

"Using a lower limit, we could find more potential front positions based on CKE, but many were far from the front positions estimated by Orsi et al (1995) and other authors (e.g., Kim and Orsi, 2014; Freeman and Lovenduski, 2016a; Chapman, 2017)."

*Reviewer Comment: Line 376*
*"The mean CKE profile pictured in Figure 3 has multiple local maxima, most likely associated with the narrow jets that surround the front." Can the author comment on how the half-power point method finds one front in this latitude band while the Orsi study finds one instance of the SACCF, two instances of the PF and one instance of the SAF (= 4 fronts) at this longitude/swath. This might suggest that the CKE method presented here can alias a front to the mean latitude of any front activity/presence (within the envelope) and not be particularly representative of a physically realistic front.*

**RESPONSE**: The method does not find one front, it finds one envelope of enhanced CKE, as we have explicitly stated. While the Orsi study finds a strong meandering PF at this location (two crossings), other methods (e.g., Kim and Orsi; Freeman and Lovenduski) do not. The reviewer is correct that the CKE regions could envelope two fronts when they are very close to one another. However, the argument still holds that if those fronts are both shifting south (as, for example, the contour method proposes), then the CKE envelope should still shift south. We have added some discussion of this in the revised paper, following the sentence quoted in the comment:

"The mean CKE profile pictured in Figure 3 has multiple local maxima, most likely associated with variability of the narrow jets that surround the front. They may also represent two separate fronts (and frontal-related jets) that are close in space. Some frontal climatologies find the SAF and PF are separated by fewer than 100 km in the South Indian Ocean (between 30°E and 40°), the South Pacific (between 220°E and 230°E), and the South Atlantic (310°E and 330°E) (Figure 2). CKE computed in these areas may encompass energy around both fronts. However, if the fronts have both shifted south (as reported in some studies), then CKE should also shift south and so tracking CKE should observe the shifts in frontal location."

*Reviewer Comment: Lines 375-386*
*While I appreciate the author's attempt to provide more information for this particular pass (via a modified figure to show variability and states the temporal range in number of defined maxima), this is for one distinct pass of the Indian sector. I conclude that no substantial quantitative information has been provided in the results in this iteration. Is there really no other quantitative information that the author can provide for this work? For example, mean and standard deviation in number of maxima or width of envelope, etc. One pass surely cannot represent the full Southern Ocean.*

**RESPONSE**: This information has been added to the Results and Analysis section, where we feel they are more relevant.

*Reviewer Comment: Lines 375-376, re: Figure 3*
*It would be helpful for clarity to add to the end of this sentence: "The mean CKE profile pictured in Figure 3 has multiple local maxima, most likely associated with variability of the narrow jets that surround the front (defined here as the location of the half-power point)."*

**RESPONSE**: This clarification is incorrect, as it suggests that the variability of the specific fronts is indicated by the half-power point. We precisely define the half-power point later, and so adding this parenthetical would add more confusion. Since the discussion at this point is only about the envelope of CKE and the various local peaks, we feel it is misleading to bring in a discussion of the half-power point at this spot.

*Reviewer Comment: Lines 377-378*
*I assume these percentages are pulled from the Chapman (2017) study - could the author place their work into context? - otherwise this feels out of place/unnecessary.*

**RESPONSE**: Thanks for pointing this out. It is out of place, and this is discussed later in relation to Figure 7. We have deleted these sentences and feel the revised paragraph reads better.

*Reviewer Comment: Lines 379-380*
*What might this suggest about the front field?*

**RESPONSE**: This discussion is on the presence of local peaks in CKE related to the highly-variable jet field. Even in the presence of a stationary front, the jets can still be highly variable due to non-linear interactions. We have added a statement here to emphasize this:

"Note that even with a fixed and stationary front, there may be highly variable locations of peaks in CKE around the front, due to the meandering and disappearance/formation of jets (e.g., Chapman, 2017a). Thus, tracking the specific jet locations is not an optimal method of tracking frontal shifts."

*Reviewer Comment: Lines 389-390*
*So again, this study does not find individual fronts (even if speaking in a mean sense) but a mean latitude of frontal activity (especially as these envelopes can sometimes span ~6 deg. latitude, which is much greater than the 2-3 degree standard deviation cited).*

**RESPONSE**: Again, we do not state (nor mean to imply) we have isolated the specific front, just the general area. We have revised the last sentence to reiterate this point yet again:

"The assumption we make in doing this is that the localized maxima are associated with variable jets, but the position of the envelope of high CKE is related to the general position of the front, and that if the front has systematically shifted then the CKE envelope will have shifted as well."

*Reviewer Comment: Lines 390-420*
*Please be clear on how the half-power point is distinct from just a sort of 'mean latitude' of the latitude band encompassing any 'ACC activity' (the full enhanced CKE envelope being the 'ACC activity' region) — and therefore the distinction between tracking actual fronts and not just a sort of 'mean ACC position' like Gille 2014 (if in fact this study is more like the latter, fine, this just needs to be clearly articulated to the reader).*

**RESPONSE**: I have added a comment to describe the differences:

"The assumption we make in doing this is that the localized maxima are associated with variable jets, but the position of the envelope of high CKE is related to the general position of the front, and that if the front has systematically shifted then the CKE envelope will have shifted as well. Other studies have tracked the mean latitude of the integrated transport computed between dynamic height contours that are picked to represent the southern boundary and the northern boundary that encompass all the fronts in the ACC (Gille, 2014). One issue with this approach is how to uniquely determine the northern and southern boundary contours without potentially biasing the result (e.g., using a priori fixed boundaries and ignoring they might have shifted). The method we propose will determines the boundaries of the integration uniquely for each pass based solely on the level of CKE relative to the peak of the enhanced CKE envelope. Moreover, it allows for two or more distinct CKE envelopes along each pass (i.e., related to different fronts), whereas the Gille (2014) method can only compute one mean latitude for all fronts in the between the prescribed southern and northern boundaries. Thus, our method is more flexible in determining boundaries around any particular front, provided the orientation of the groundtrack is such that the majority of jets are perpendicular to it."

*Reviewer Comment: page C14, paragraph 2*
*1. As hoped, the addition of Figures 6 and 7 have not yet alleviated my previous concerns about actual front identification. This stems from most of my other comments here on actual frontal detection vs. a mean ACC position.*

**RESPONSE**: I hope the several additional statements (discussed previously) noting we are not detecting specific fronts, but using shifts in the CKE envelope as a proxy for detecting shifts in the fronts will alleviate these concerns.

*Reviewer Comment: 2. I don't believe the addition of Figure R2 to the manuscript is necessary. As I understand it, the Chambers 'front' would then be defined as the half-power point (not plotted here but) located to the south of the other studies' fronts (between 50 and 51S; same as red dot in Figure 3)? If thinking in terms of actual named fronts, the Chambers front detection method would find an equivalent PF at this particular location, not the SAF. And when comparing to Figure 7, this appears to be the mean latitude of the colored variability area.*

**RESPONSE**: Thanks for confirming the additional figure is not necessary. All we attempt to say is that the true front should be somewhere around the half-power point, within the envelope. The important thing is that if the front has shifted, this envelope (and half-power point) should also shift. We have stated this several times already and don't believe it is necessary to state it once again here.

*Reviewer Comment: page C3, paragraph 3*
*Thank you for testing this. I don't think that including the one sentence you provided as a response will take up too much space in the manuscript and will only benefit transparency: 'We find that mean front positions are not sensitive to choice of x-year periods.'*

**RESPONSE**: The following comment was added at the end of the section, noting that we focus on the shift in the half-power point, not it's specific location:

"We tested different averaging periods (ranging from 1- to 4-years), but found the estimate in overall shift of the half-power point over the 24-year period was insensitive to the choice."

*Reviewer Comment: Figure 6*
*1. I understand that the author added Figure 6 to boost the argument for/robustness of the presented methodology (given that the mean half-power point lies near previous climatologies) but it does not necessarily confirm that the methodology is actually locating fronts at a given time. As in the first review, I still feel strongly that the text of the manuscript misrepresents/misstates what is actually being performed.*

**RESPONSE**: We have revised the manuscript in several places to make it clear we are not trying to detect exact locations of fronts with the method, merely identifying envelopes and tracking how they have shifted. Showing the mean-half power points along with front climatologies indicates the locations are reasonably near those estimates (and often between two different estimates) giving confidence the estimates are detecting regions of high CKE related to jets around fronts, not wind-driven mesoscale eddies.

This is discussed in the manuscript shortly after the comment:

"The comparison between CKE half-power points and front climatolgies is reassuring that the method developed in Section 2 is successfully detecting regions of high energy related to jets around fronts. Since the movement of jet positions has been used to estimate movement of the fronts (e.g., Chapman, 2017a), a comparable calculation with positions of high CKE seems reasonable."

*Reviewer Comment: 2. It is very difficult to decipher individual fronts within the same color grouping (particularly the orange and blue groupings), font sizes are too small throughout, and the overall quality/resolution of the figure is low.*

**RESPONSE**: I apologize for this. The original figure resolution is > 500 dpi, but it was reduced in the PDF creation. I have increased fonts for this figure and made some changes to the colors and line types to hopefully alleviate this problem.

*Reviewer Comment: 3. Since the author claims that envelopes are being tracked, doesn't this suggest that some sort of envelope should also be indicated around your mean CKE dots as well? Single dots hide the fact that there is a known width (meridional extent N-S where CKE exceeds 200 units) that is unique to each location and time step (e.g., something like the spread in the colors plotted in Figure 7). It would help quantitatively to provide either a mean or standard deviation in the N-S width of the envelopes surrounding the mean CKE latitudes. And then going further, any detected shifts could be interpreted in light of this underlying/inherent envelope width.*

**RESPONSE**: I created a version of the figure with the envelopes, but it made an already complicated figure even more problematic. Instead, I discuss the sizes of the envelopes relative to the approximate distance between fronts in Section 3, as discussed in a previous response.

*Reviewer Comment: 4. Please comment on the result that most of the CKE method findings would appear to be the SAF, as revealed in Figure 6. I know I asked this previously but this same line of thought came up again this round. And I honestly don't know what to make of the author's response on page C15, second to last P. If anything, this confirms that this study is detecting an ACC feature but with no information on which feature unless you relate/compare to previous studies/climatologies.*

**RESPONSE**: We have added a comment about this before discussing Figure 7.

"The majority of the estimated half-power points follow the SAF. This is most likely due to the front (and jets) moving perpendicular to the groundtracks along this front. This method will tend to only detect high CKE when the front is moving from northwest-to-southeast for an ascending pass, and from southwest-to-northeast for a descending pass. This method also only works in regions where the front is associated with highly variable jets, which does not occur at every longitude along the front (e.g., Chapman, 2017a)."

*Reviewer Comment: Figure 7*

*Very difficult to decipher colored dots; perhaps make this a 4-panel figure with x-y plots allowing for zooming in on the frontal area (e.g., limiting the latitude range of each)?*

**RESPONSE**: I do not understand your idea about changing the figures. I tried to make this approximately the same latitude range as Figure 6 for easy comparison. I have made the back dots indicating the half-power point larger.

*Reviewer Comment: Line 546*
*I disagree with the implication/tone of this sentence. The fact that the various studies' climatologies don't lie on top of one another is not that the calculations that went into finding the fronts are uncertain, but is a result of different methodologies used, which includes different time periods and different data sets. Each front analysis should be treated and interpreted based on its own methodology. A caveat is different from an uncertainty.*

**RESPONSE**: The level of disagreement between different estimates is one measure of uncertainty. Such a calculation is often done to quantify uncertainty in different climate models, for instance. If all methods found the same location, one would have a greater confidence in the ability to detect fronts in the Southern Ocean.

However, we have tried to change the tone by revising the paragraph:

"It is important to note the large differences in estimates for the same front, which indicates how difficult it is to determine fronts in a highly variable current system like the ACC. For instance, in the Indian Ocean at 50°E, Freeman and Lovenduski (2016a) find the PF at the same location that Orsi et al. (1995) found the SAF, while Kim and Orsi (2014) find it significantly farther south. The SAF determination using the contour method (Kim and Orsi, 2014) is substantially farther north than the one determined from hydrographic data (Orsi et al., 1995) at most longitudes. These differences are likely due to differences in the time-span, differences in methodologies, and uncertainty in the data utilized. All lead to a level of uncertainty in the determination of a specific front at any time."

*Reviewer Comment: Line 547*
*Please replace the period with a colon (prior to 'For instance,…') to make clear that the author is discussing the spread in the various estimates across the different studies.*

**RESPONSE**: This was not done. As revised, that would lead to a very long sentence. We prefer breaking into two sentences.

*Reviewer Comment: Lines 552-553*
*This sentence is muddy. Does the author mean, "Many front location estimates, defined as the half-power points of enhanced CKE, are found within the spread of the PF or SAF across multiple studies." ?*

**RESPONSE**: The sentence has been revised to:

"The half-power points of enhanced CKE generally occur near or between the fronts estimated by different methods (i.e., the three different PF estimates), indicating they are at least within the uncertainty bounds of frontal detection by other methods."

*Reviewer Comment: Lines 554-557*

*I'm not sure this comparison is fair and appropriate. This study has not claimed to find the PF in particular, and in fact, most half-power points seem to lie along the SAF (as determined by using previous climatologies to compare qualitatively); we have not been provided any information on the standard deviation of the SAF. As a reader, I feel misguided here.*

**RESPONSE**: I was using the one published estimate of variability of a front as a proxy for variability of others. I have made this more clear in the revision by adding:

"Some values are at locations either north or south of the other front estimates by as much as 3°, but it should be noted that the standard deviation of the PF estimated by Freeman and Lovenduski (2016a,b) averages 2-3°. Using PF variability statistic an indicator of variability of all fronts, one can conclude the location CKE half-power points are well within the level of expected frontal variability and so not statistically too distant from a front location."

*Reviewer Comment: Line 558*
*'Probably a better method…' ??? The author presents the methodology but then states that a different method presented by another study (in this case, Chapman, 2017) is superior. Please address.*

**RESPONSE**: I meant for detecting a mean front position, which this study does not attempt to do. However, to avoid confusion, I have changed the beginning to:

"Another method for determining frontal position is to examine the probability of jets occurring."

*Reviewer Comment: Lines 565-567*
*Since this is the only discrepancy that the author points out, it warrants further discussion. Please comment. Personally, I'm curious whether this is an artifact of the half power-point methodology - it is curious that the half-power point is found between two regions to the north and south where Chapman would 'probably' find a feature (colored in this figure) and likely matches the north and south bounds of an 'envelope' identified by the author. Has the author isolated this particular region/swath and looked at the features of this envelope (like Figure 3)? Is the half-power point aliasing the front between two major peaks here?*

**RESPONSE**: Although I do not feel discussing two points out of 150 is relevant, I have looked at this and provide a short commentary in the revised paper. First, one has to remember the groundtrack swath is not directly south-to-north, but more northwest-to-southeast (**Figure R1**). Thus, we should compare those values of the Chapman database.

Second, in these two regions, Chapman only finds a jet less than 10% of the time. If one looks at the mean CKE envelopes (**Figure R2**), Chapman finds jets on the northern side of the envelope only, and not in the center (where the half-power point is by definition) and not where the CKE has a peak. We can't explain why Chapman finds jets south of the peak CKE, as the CKE is quite low there. This could be due to either the orientation of the groundtrack with the jets, a problem in the gridded data Chapman used, or an error in Chapman's method. Confirming which is beyond the scope of this paper, especially since it only occurs with approximately 1% of the comparisons.

Based on these figures we conclude there is no aliasing of CKE half-power point between two peaks of enhanced CKE. The method works exactly as intended, finding a value in the area of highest CKE.

Our brief statement on this is:

"However, it should be noted that Chapman finds jets in the two areas north and south of these two CKE half-power points less than 10% of the time and that the northern cluster lies on the northern edge of the enhanced CKE envelope. Although the half-power points are slightly south of this along these two passes, this is due to high CKE (in excess of 200 cm$^2$ s$^{-2}$) down to 58°S, where Chapman (2017a) detects few jets. It is unclear why Chapman (2017a) detects few jets in this region of high CKE, but it should be noted that this represents only 1% of the samples compared."

We feel this should be sufficient to answer the reviewer's concerns without adding the figures or a longer discussion.

[Figure]

**Figure R1**. Groundtrack of two passes where CKE half-point does not align with probability of a jet in the Chapman database (red dots in Figure). The Black circles indicate regions where there are a relatively large number of jet locations, while the solid black dots indicate where the mean CKE half-power point was found.

[Figure]

**Figure R2**. Mean CKE along the two passes shown in Figure R1, along with the CKE half-power point (solid squares) and the mid-point of the northern Chapman jets (open squares).

*Reviewer Comment: Lines 568-571*
*Please update/remove casual/informal language like 'the good comparison' here.*

**RESPONSE**: This was changed to:

"The comparison between CKE half-power points and front climatolgies is reassuring…."

*Reviewer Comment: Lines 635-636*
*I'm not sure this claim can be made. The author makes no distinction between the PF and SAF throughout the manuscript but here, focuses on shifts in this study's SAF specifically. Please address.*

**RESPONSE**: The reviewer has mentioned several times in this second review that the CKE half-power points cluster around the SAF and has specifically asked us to comment on this: "…*and in fact, most half-power points seem to lie along the SAF*" and "*Please comment on the result that most of the CKE method findings would appear to be the SAF*". The results being discussed here (from Kim and Orsi) only examined the SAF. Since our estimates cluster around the SAF in the same region Kim and Orsi discussed, we feel it is fair to comment on the similarities and differences.

We have, however, modified the sentence slightly to read:

"We also find some locations in this region, where the CKE half-power points cluster around the SAF, also have a significant northward shift."

*Reviewer Comment: Figure 8*
*This study tracks the half-power points of the envelope of enhanced CKE. Therefore, they inherently have an underlying width. If we had information on the width of the CKE envelopes (i.e., ACC activity), we might also be able to confirm whether these shifts are significant given the widths (i.e., does the confidence interval associated with the change in the single point exceed the width and variability of the underlying envelope?). I guess that follows from whether the author could provide a sort of error statistic that relates the half-power point to the associated width of each envelope.*

**RESPONSE**: The comparison of the width of the envelope to movement of the half-power point is a meaningless statistic in this case – the width will ALWAYS be significantly larger the movement of the half-power point. Examining shifts in the mean of a distribution (i.e., the half-power point in this case) is a perfectly valid and widely accepted statistic. The uncertainty is then computed based on the standard error in the determination of the mean or the variability of the mean. Only in rare cases is the standard error in the mean determination larger than the intrinsic variability. In this case, the standard error in the determination of the half-power point is of the order of 10 km, based on the sampling of the data and the failure to find a point where the integrals exactly match. Yet the standard deviation of the half-power points around the mean and the fit are much greater than this, so they drive the uncertainty in the estimated trend.

All of this is accounted for in the uncertainty bars and is described fully at the end of Section 3.

Thus, we will not add the requested statistics here in the conclusions, as they are not relevant.

[revised manuscript text omitted]

---

## Author Response (AR3)

Matthew Hecht
Editor, *Ocean Science*

Dear Dr. Hecht,

I have made the minor changes you indicated in your acceptance letter and have uploaded all relevant files. Thanks again for your attention in editing this paper.

Cheers,

Don Chambers